# Pre-Trained Policy Discriminators are General Reward Models

Shihan Dou[1,2*‡], Shichun Liu[1,2*‡], Yuming Yang[1,2*], Yicheng Zou[1*†],
Yunhua Zhou[1], Shuhao Xing[1], Chenhao Huang[2], Qiming Ge[1], Demin Song[1], Haijun Lv[1],
Songyang Gao[1], Chengqi Lv[1], Enyu Zhou[2], Honglin Guo[2], Zhiheng Xi[2], Wenwei Zhang[1],
Qipeng Guo[1], Qi Zhang[2], Xipeng Qiu[2], Xuanjing Huang[2], Tao Gui[2†], Kai Chen[1†]

[1]**Shanghai AI Laboratory,**    [2]**Fudan University**
{zouyicheng,chenkai}@pjlab.org.cn, tgui@fudan.edu.cn

 https://github.com/InternLM/POLAR

## Abstract

We offer a novel perspective on reward modeling by formulating it as a policy discriminator, which quantifies the difference between two policies to generate a reward signal, guiding the training policy towards a target policy with desired behaviors. Based on this conceptual insight, we propose a scalable pre-training method named **POL**icy Discrimin**A**tive Lea**R**ning (**POLAR**), which trains a reward model (RM) to discern identical policies and discriminate different ones. Unlike traditional reward modeling methods relying on absolute preferences, POLAR captures the relative difference between one policy and an arbitrary target policy, which is a scalable, high-level optimization objective suitable for modeling generic ranking relationships. Leveraging the POLAR pre-training paradigm, we present a series of RMs with parameter scales from 1.8B to 7B. Empirical results show that POLAR substantially outperforms traditional non-pre-trained methods, significantly enhancing RM performance. For instance, POLAR-7B could improve preference accuracy from 54.8% to 81.0% on STEM tasks and from 57.9% to 85.5% on creative writing tasks compared to SOTA baselines. POLAR also shows robust generalization capabilities in RLHF using Reinforcement Fine-tuning (RFT), providing reliable reward signals and markedly enhancing policy performance—improving LLaMa3.1-8B from an average of 47.36% to 56.33% and Qwen2.5-32B from 64.49% to 70.47% on 20 benchmarks. Moreover, scaling experiments reveal a clear power-law relationship between computation and performance, supported by linear correlation coefficients approaching 0.99. The impressive performance, strong generalization, and scaling properties suggest that POLAR is a promising direction for developing general and strong reward models.

## 1 Introduction

Reinforcement learning (RL) plays a crucial role in the post-training of large language models (LLMs) [129; 80; 5]. Its success hinges on the reward model's (RM) ability to provide precise and stable feedback to the policy model [110; 29]. Although recent approaches successfully leverage labeled preference pairs to train RMs for alignment with human preferences, these methods often face challenges in terms of scalability and generalization [119; 66; 74; 51; 108]. The former is limited by the difficulty of acquiring large volumes of high-quality labeled pairs [24; 22], while the latter stems from the fact that this subjective approach to modeling human preferences makes RMs vulnerable to reward hacking [16; 13; 123]. On the other hand, several works, such as DeepSeek's R1 [35], utilize

---

*Equal contributions. †Corresponding authors. ‡Work done during an internship at Shanghai AI Laboratory.

39th Conference on Neural Information Processing Systems (NeurIPS 2025).

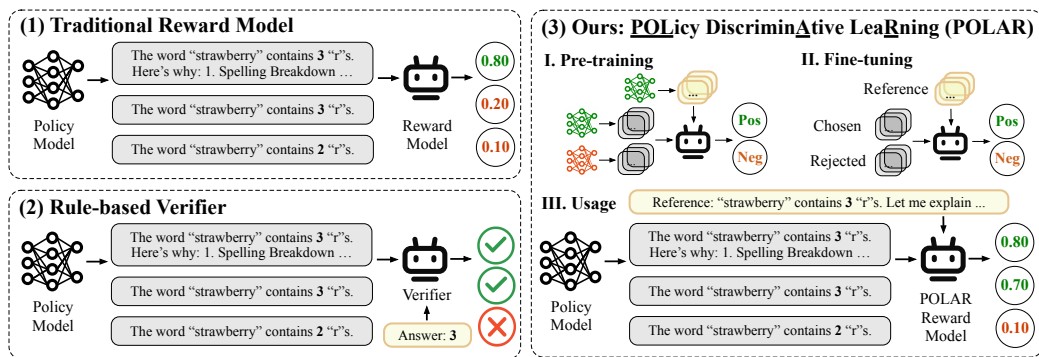

Figure 1: Comparison of three reward modeling methods: **(1)** traditional methods incorporate absolute preferences into RMs, which directly assess the quality of trajectories; **(2)** rule-based verifier validate the candidate trajectory through the gold answer and predefined rules; **(3)** POLAR pre-trains an RM to recognize identical policies and discriminate different ones, enabling it to measure the difference in trajectories between a training policy and a target policy with desired behaviors.

rules to verify the correctness of model outputs and provide accurate reward signals for RL. However, these rule-based verifiers can only be applied in scenarios where model outputs can be automatically verified by pre-defined rules, such as in reasoning and coding tasks [31; 113; 65; 83]. In open-ended domains like writing and translation, rules are usually complicated and difficult to design in advance, making the rule-based verifiers hard to extend to general tasks.

Before delving into reward modeling, it is instructive to revisit the widespread success of LLMs. By adopting a unified Next Token Prediction (NTP) optimization target [87; 88; 10], LLMs effectively harmonize diverse NLP tasks under a common objective, addressing the challenge of cross-task generalization. This inspires us to reconsider the training paradigm of RMs. Traditional RMs heavily rely on absolute, manually-defined criteria to generate a preference score. Analogous to how LLMs unified NLP tasks, we should identify a fundamental, criterion-agnostic objective for RM pre-training.

Instead of traditional absolute preference modeling, we propose redefining a reward model as a "policy discriminator". Specifically, by quantifying the difference between candidate policies and a given target policy, we establish a criterion-agnostic objective, which naturally assigns higher scores to policies that are more "similar" to the desired target policy. This reward signal could guide the training policy toward desired behaviors during RL. Furthermore, since target policies can be arbitrarily chosen, this objective eliminates reliance on manually defined preferences and is applicable to any scenario, thus offering a scalable and fundamental pre-training paradigm for RMs. We refer to this training objective as Policy Discriminative Learning (**POLAR**), as illustrated in Figure 1.

Starting from a diverse and extensive collection of policy models, we construct a large-scale synthetic corpus by sampling trajectories from these policies. We then formulate the pre-training task as a contrastive learning objective utilizing Bradley-Terry (BT) loss [9], which encourages the RM to recognize trajectories derived from identical policies, while distinguishing those originating from different ones. Consequently, the pre-trained RM learns to assign higher rewards to trajectories exhibiting greater consistency with the target policy and generalizes this discrimination capability to unseen policies. After pre-training, analogous to supervised fine-tuning (SFT) in LLMs [80; 87] that enables rapid adaptation to specific tasks and instructions, we also introduce an SFT procedure for POLAR RMs tailored to align with human-defined criteria. This fine-tuning process requires only a small set of reference trajectories generated from a given target policy, accompanied by candidate trajectories annotated with ranking labels reflecting their difference relative to the target. The reference trajectories can be directly annotated by humans, or alternatively, they could be derived from high-performing LLMs. Such a flexible and lightweight fine-tuning approach allows the RM to rapidly adapt to new domains or criteria.

Leveraging the POLAR pre-training method, we present a series of reward models with parameter scales ranging from 1.8 to 7 billion. Our empirical results demonstrate that POLAR effectively discriminates among diverse policies, and the fine-tuned POLAR RMs could significantly outperform traditional preference modeling methods. Specifically, in preference evaluation tasks, POLAR-7B surpasses the SOTA 72B-parameter WorldPM [111], achieving an average improvement of 5.8%

points despite being approximately $10\times$ smaller. Additionally, when applied within RLHF using Reinforcement Fine-tuning (RFT) [70; 79], POLAR RMs deliver more accurate reward signals and exhibit superior generalization across various downstream tasks, substantially enhancing the performance of popular policy models such as Qwen2.5 [120], LLaMa3.1 [27], and InternLM3 [12]. Moreover, POLAR exhibits scaling laws similar to those observed in LLMs, highlighting its significant potential for developing increasingly powerful reward models. In summary, our main contributions are as follows:

1. We propose POLAR, a novel criterion-agnostic pre-training paradigm for reward modeling based on a scalable training objective—policy discrimination.

2. Scaling experiments reveal promising scaling laws, highlighting the significant potential of POLAR for enhancing the upper bound of reward modeling and developing stronger and more generalizable RMs.

3. We developed the POLAR series of reward models. They substantially outperform traditional RMs in empirical evaluations, achieving higher preference accuracy and better generalization than considerably larger RMs. This advancement expands the potential and applicability of RL algorithms, such as RFT, paving the way for more innovative and diverse applications.

## 2 Related Work

**Reward Modeling in Reinforcement Learning** RL has emerged as a pivotal technique in the post-training phase of LLMs [54; 52; 109; 5; 84; 35; 80; 129; 124]. All RL approaches crucially depend on precise reward signals provided by the reward function. Reward functions broadly fall into model-based and rule-based categories, with our work aligning closely with the model-based reward function paradigm [29; 109; 53]. Existing model-based methods typically train RMs using labeled pairs to approximate a preference distribution [5; 92; 111]. However, the scarcity of extensive labeled preference pairs poses a significant challenge [24; 22; 6; 45]. Additionally, such methods often exhibit limited generalization, struggling to predict out-of-distribution preferences robustly, thus weakening their effectiveness in RL training [16; 13; 123]. Another line of research investigates generative reward models, leveraging LLMs themselves as verifiers [128; 44; 72; 126; 68; 15; 36]. Yet, these approaches inherently reflect the biases and preferences of the models used, frequently resulting in unreliable or biased reward signals [34]. Alternatively, some studies utilize rule-based verifiers to provide feedback [115; 65; 35; 105], but this approach struggles to address general tasks that are challenging to verify through predefined rules.

**Pre-training and Scaling Laws** Our work closely relates to unsupervised learning, a critical component of pre-training focused on deriving essential and generalizable knowledge from extensive unlabeled datasets [82; 10; 88; 37; 81; 21; 91; 87]. Unsupervised pre-training is foundational to modern advances in language modeling, powering the latest sophisticated AI systems [86; 25; 114; 56; 112; 69]. Concurrently, scaling laws describe the power-law relationship among computation and performance in neural language models, offering crucial insights into unsupervised pre-training [50; 11; 46; 40; 95; 117; 8]. Extensive research indicates that scaling up data size and model size consistently enhances LLM performance [27; 20; 85; 76; 4; 38; 87; 50; 127]. Recent studies further demonstrate the capability of scaling laws to predict the performance of larger models based on smaller ones, enabling efficient resource allocation [95; 46; 40]. The success of scaling laws in language modeling has significantly influenced theoretical understandings and practical advancements, inspiring similar explorations in vision [89; 97; 101; 64], multimodal learning [64; 1; 23; 30; 107], and reinforcement learning [41; 75]. Gao et al. [29] specifically expanded scaling laws into the over-optimization of preference-based reward models, further broadening their applicability in RMs.

**Discriminator-based Reward Modeling** Generative Adversarial Imitation Learning (GAIL) [42] and Discriminative Reward Co-Training (DIRECT) [2] are two representative works of discriminator-based reward modeling. Both approaches train a discriminator to distinguish between policy-generated trajectories and high-quality ones, and then use the discriminator's output as a reward signal. These methods are mainly designed for single-task scenarios, where the objective is to imitate expert behavior or reproduce past successes. Our work differs from these approaches and benefits from large-scale pre-training. Instead of relying on expert demonstrations, we pre-train a policy discriminator that learns criterion-agnostic differences between policies from a large-scale unsupervised dataset.

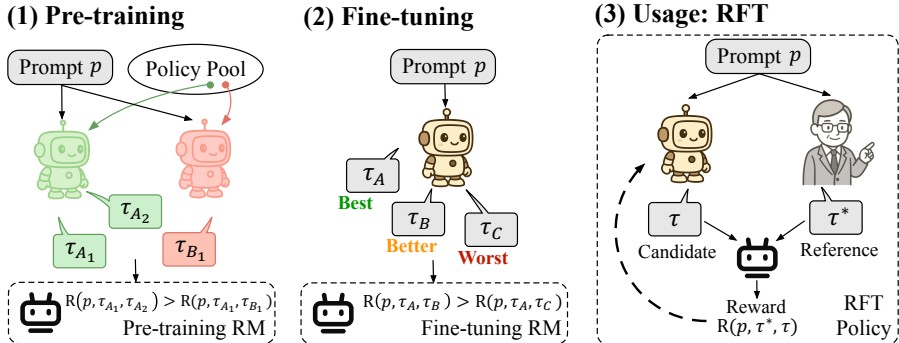

Figure 2: Overview of Policy Discriminative Learning (POLAR). **Stage 1:** In pre-training, the RM learns criterion-agnostic policy differences by assigning higher rewards to trajectory pairs from consistent policies. **Stage 2:** During fine-tuning, human annotators rank trajectories from the same policy, implicitly defining human criteria, to align RM evaluations with human standards. **Usage:** In Reinforcement Fine-Tuning (RFT), the fine-tuned RM provides reward signals comparing candidate trajectories with human-preferred references, guiding policy training toward desired behaviors.

## 3 Method

In this section, we first revisit the optimization objectives of RL in LLMs and show that the RM essentially functions as a policy discriminator. Building on this insight, we introduce the Policy Discriminative Learning (POLAR) as illustrated in Figure 2. During the first stage, the RM acquires the ability to discriminate among various policies and quantify their differences. Subsequently, in the supervised fine-tuning stage, we fine-tune the RM on trajectory ranking data derived from human judgments, explicitly capturing preferences regarding policy behaviors.

### 3.1 Preliminary #1: Preference-based Reward Models and Their Limitations

In the RLHF pipeline, reward models are typically trained with pairwise preference data. Given a prompt and two candidate responses, human annotators are asked to choose which response better aligns with their preferences. The RM is then optimized to assign a higher score to the preferred response, often using a Bradley–Terry (BT) loss [9] or an equivalent ranking objective. Formally, for a prompt $x$ and two responses $\tau_A$ and $\tau_B$, the RM $r_\theta$ is trained to satisfy

$$P(\tau_A \succ \tau_B \mid x) = \sigma\big(r_\theta(x, \tau_A) - r_\theta(x, \tau_B)\big), \tag{1}$$

where $\sigma(\cdot)$ is the logistic function. This framework allows the RM to approximate a latent reward function consistent with observed human preferences. We refer to this approach as **absolute preference modeling**, since it explicitly encodes what types of responses are judged as good or bad.

Despite its effectiveness in many alignment tasks, preference-based training is fundamentally constrained by its reliance on subjective human judgments. Annotation standards may vary across individuals, leading to bias and inconsistency in the preference labels. The collection of large-scale, high-quality data is also expensive and time-consuming, which limits its scalability. Moreover, the resulting models often exhibit weak generalization, struggling to extrapolate beyond the distribution of training data. These limitations suggest that absolute preference modeling may not be suitable for robust alignment. This motivates a shift toward exploring alternatives and more scalable paradigms.

### 3.2 Preliminary #2: Policy Optimization in Reinforcement Learning

Consider a training policy model $\pi_\phi$ parameterized by $\phi$ and a prompt distribution $\mathcal{D}_x$. Let $\tau$ denote a trajectory generated by this policy given prompts sampled from $\mathcal{D}_x$. The resulting policy distribution $\mathcal{D}_{\pi_\phi}$ can thus be empirically approximated by sampling prompt-trajectory pairs $(x, \tau)$. Let $r_\theta : (x, \tau) \to \mathbb{R}$ denote the reward model parameterized by $\theta$, which assesses policy performance by assigning scores to prompt-trajectory pairs $(x, \tau)$. Given a regularization coefficient $\beta$ that controls the strength of the KL-divergence penalty relative to the initial policy $\pi_{\text{init}}$, the RL optimization objective can be formulated as follows [80]:

$$\mathcal{O}_{\text{RL}}(\pi) = \mathbb{E}_{(x,\tau)\sim\mathcal{D}_\pi}\left[r_\theta(x, \tau) - \beta D_{\text{KL}}(\pi(\tau|x)||\pi_{\text{init}}(\tau|x))\right]. \tag{2}$$

The optimal policy $\pi^*$ then admits a closed-form solution [90]:

$$\pi^*(\tau|x) = \frac{1}{Z(x)}\pi_{\text{init}}(\tau|x)\exp\left(\frac{r_\theta(x,\tau)}{\beta}\right),\tag{3}$$

where $Z(\cdot)$ is the partition function. This formulation provides a key insight: the reward model implicitly encodes a continuous operator that maps an initial policy to its optimal form through RL optimization, captured by the reward scores. Consequently, when a policy $\pi_\phi$ is trained against this reward signal under KL constraints, it effectively learns to approximate this implicit mapping within the defined reward distribution.

### 3.3 Unsupervised Reward Pre-training via Distributional Alignment

Traditional reward modeling approaches are based on explicit pairwise comparisons (e.g., harmlessness in safety alignment), which inherently presuppose an absolute human criterion. However, this assumption becomes problematic when considering broader, more general classes of criteria. Specifically, given an arbitrary criterion $p$ inducing a partial ordering over responses, i.e., $\{r(a_1) \geq r(a_2) \geq \cdots \geq r(a_n)\}$ for responses $a_i$, there often exists a complementary criterion $\neg p$ that reverses this ordering. Consequently, exhaustively enumerating all possible criteria is not only computationally infeasible but also theoretically ill-posed.

To overcome this challenge, we propose an unsupervised reward pre-training paradigm that provides a criterion-agnostic initialization for the reward model. Specifically, let $\pi^*$ and $\pi_{\text{init}}$ represent two policy distributions, where $\pi^*$ denotes the optimal policy for a given downstream task and $\pi_{\text{init}}$ denotes the initial policy. Under the KL-constrained RL objective $\mathcal{O}_{\text{RL}}$, we observe that the reward function can be uniquely defined (up to an additive constant) by the density ratio between $\pi^*$ and $\pi_{\text{init}}$, resulting in the following relationship:

$$r_\theta(x,\tau) \stackrel{\triangle}{=} \beta\log\frac{\pi^*(\tau|x)}{\pi_{\text{init}}(\tau|x)} + \beta\log Z(x).\tag{4}$$

Since the partition function is a constant independent of trajectories, we focus only on the former item, whose expected reward spans the entire trajectory space $\tau$:

$$\mathbb{E}_{\tau\sim\pi^*(\cdot|x)}\left[r_\theta(x,\tau)\right] \sim \beta\underbrace{\mathbb{E}_{\pi^*}\left[\log\frac{\pi^*}{\pi_{\text{init}}}\right]}_{D_{\text{KL}}(\pi^*\|\pi_{\text{init}})}.\tag{5}$$

This term converges to zero when the sampling distribution precisely matches the optimal distribution. Motivated by this observation, we replace direct reward regression with a distributional distance minimization approach. Given prior knowledge of desired behaviors $\tau^*$ (e.g., human references), we train $r_\theta$ to measure the divergence between $\pi^*$ (approximated through $\tau^*$) and $\pi_{\text{init}}$ (approximated through policy rollouts).

Under this perspective, the role of the RM fundamentally shifts: instead of merely assessing the performance of individual trajectories, it now serves as a measure of policy differentiation, quantifying differences between the training policy and the desired target policy. In practice, given two different trajectories generated from the same prompt, the pre-trained RM estimates the divergence between their underlying sampling policies, which can be formalized as follows:

$$D(\pi_\phi,\pi^*) = \mathbb{E}_{(x,\tau)\sim\mathcal{D}_{\pi_\phi},(x,\tau^*)\sim\mathcal{D}_{\pi^*}}\left[r_\theta(\tau,\tau^*|x)\right].\tag{6}$$

This perspective naturally suggests viewing the reward model as a **policy discriminator**: it learns to distinguish between the training and target policies, quantifying their degree of difference. A smaller difference yields a larger assigned reward, thus incentivizing the training policy to progressively align more closely with the desired target policy through RL optimization. Here, we employ Bradley-Terry (BT) loss [9; 100] to pre-train the reward model:

$$\mathcal{L}_{\text{pre-train}}(\theta) = -\mathbb{E}_{(p,\tau_{A_1},\tau_{A_2},\tau_{B_1})\sim\mathcal{D}_{\text{pre-train}}}\left[\log\sigma\left(r_\theta(p,\tau_{A_1},\tau_{A_2}) - r_\theta(p,\tau_{A_1},\tau_{B_1})\right)\right].\tag{7}$$

Here $p$ denotes a prompt sampled from the pre-training dataset $\mathcal{D}_{\text{pre-train}}$. $\sigma$ denotes the sigmoid function. $\tau_{A_1}$ and $\tau_{A_2}$ are generated by the same policy, while $\tau_{B_1}$ originates from a different policy. These policies are randomly selected from a diverse policy pool comprising LLMs varying in architectures and parameters. The pre-training objective encourages the RM to capture nuanced distinctions between policies, assigning higher rewards to trajectory pairs drawn from more closely aligned or "similar" policies.

## 3.4 Supervised Fine-tuning with Human Criteria

After unsupervised pre-training, the reward model acquires a criterion-agnostic capability to discriminate between different policies. For practical usage, it is essential to align this discriminative ability with human-defined standards and preferences. To this end, we introduce a supervised fine-tuning stage, which is designed to efficiently adapt the pre-trained RM to human judgment criteria.

Ideally, this fine-tuning stage would utilize reference trajectories generated by a desired target policy, accompanied by candidate trajectories annotated with ranking labels indicating their relative differences. In practice, however, we adopt a simplified yet effective approach: given a prompt $p$ from downstream tasks, we generate three distinct trajectories from the same policy. Human annotators then rank these trajectories from best to worst, denoted as $(\tau_A \succ \tau_B \succ \tau_C)$, where $\tau_A$ is the most preferred, $\tau_B$ is the second-best, and $\tau_C$ is the least preferred. Although these trajectories originate from a single policy, the introduction of human evaluations implicitly imposes different human criteria, effectively treating these trajectories as if they were drawn from different underlying policies that reflect varying degrees of alignment with human standards. During fine-tuning, we employ the following supervised ranking loss, consistent with the Bradley-Terry framework:

$$\mathcal{L}_{\text{fine-tune}}(\theta) = -\mathbb{E}_{(p,\tau_A,\tau_B,\tau_C)\sim\mathcal{D}_{\text{fine-tune}}} \left[\log \sigma \left(r_\theta(p, \tau_A, \tau_B) - r_\theta(p, \tau_A, \tau_C)\right)\right], \tag{8}$$

where $\tau_A$ denotes the trajectory ranked highest by humans and $\tau_C$ is ranked lowest. By optimizing this loss, the reward model quickly adapts to human preferences, effectively measuring the policy differences implied by human judgments, thereby closely aligning its policy discriminative capabilities with human evaluation criteria. This SFT stage effectively bridges the gap between the criterion-agnostic discrimination learned during pre-training and human-aligned evaluations, yielding a reward model robustly generalized to human judgment scenarios with minimal annotation overhead.

# 4 Experiments

We train reward models with parameter sizes of 1.8B and 7B using POLAR, denoted as POLAR-1.8B and POLAR-7B, respectively, and benchmark their performance against SOTA baseline RMs. In Section 4.1, we specify the model and training details. Section 4.2 presents the evaluation results on preference modeling benchmarks. In Section 4.3, we demonstrate the efficacy of POLAR within the RLHF framework. To further highlight POLAR's potential and scalability, we scale up the model sizes and analyze the corresponding scaling laws in Section 4.4. Lastly, Section 4.5 presents comprehensive ablation studies to validate the contributions of different training stages in POLAR.

## 4.1 Model and Training Details

**Model Architecture**  The architecture of POLAR RMs is based on an autoregressive Transformer [87; 88; 10], similar to models in the GPT series [87; 88], augmented with a linear prediction head—a common design adopted in traditional preference-based reward modeling methods [63; 12]. Specifically, the decoder configuration used in POLAR RMs aligns with that of the InternLM-2.5 series [12]. For traditional RMs, the prompt and trajectory are concatenated to form the model input, from which the model directly outputs a reward value. In contrast, as outlined in our training objective, POLAR takes a prompt along with two trajectories—a reference and a candidate—as input. We utilize special tokens to combine these three elements into a single input sequence:

prompt + reference <|split_token|> prompt + candidate <|reward_token|>

The linear head then processes the hidden state corresponding to the <|reward_token|> token from the model's final layer to produce the reward value.

**Training Details**  During the pre-training stage, we follow the standard process widely adopted in LLM pre-training [12; 62; 120] and conduct extensive experiments on hyperparameters, deriving clear scaling laws for hyperparameter configurations. Appendix D.1.2 provides an in-depth analysis of how hyperparameters are chosen for POLAR. Rather than training from scratch, we initialize the POLAR models from a pre-trained InternLM2.5-series model and perform one additional epoch of POLAR pre-training. Training details for the pre-training and supervised fine-tuning stages, including data composition, are provided in Appendices D.1 and D.2, respectively.

## 4.2 Performance in Preference Evaluation

We first evaluate POLAR on the human preference prediction task, which measures the ability of RMs to accurately identify responses preferred by humans.

**Evaluation Setup** We compare POLAR with 5 SOTA RM baselines: InternLM2-Reward-7B [12], InternLM2-Reward-20B [12], Skywork-Reward-8B [63], Skywork-Reward-27B [63], and WorldPM-72B-UltraFeedback [111] (details in Appendix E.1). Our primary evaluation uses the RMB benchmark [130], containing 3,162 questions, each with multiple trajectories ranked by preference scores. The top-ranked trajectories are treated as references, representing samples drawn from a target policy. The task is to identify whether RMs correctly prefer the second-ranked trajectory over the third-ranked one. Additionally, we create another evaluation set from real user queries collected through online platforms and manually annotate trajectory rankings (see Appendix E.2). We carefully remove overlaps with the training data to maintain independence. Existing RM baselines typically assess trajectories without considering references. To ensure fairness, we evaluate baselines using two methods: (a) standard scoring without references, and (b) including references explicitly in the prompt (Figure 10). We report the best results for each baseline across these two settings.

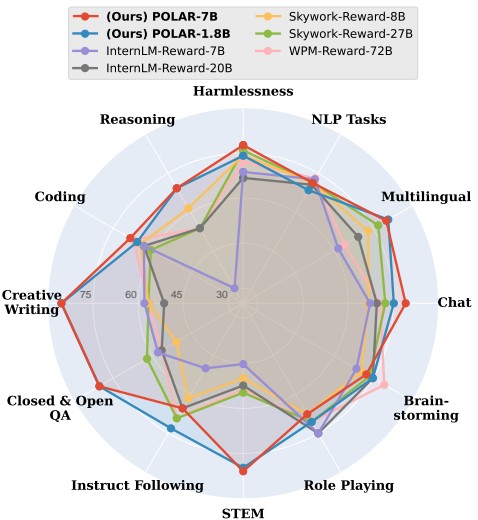

Figure 3: Comparison of POLAR and baselines on human preference prediction.

**Results Comparison** To thoroughly illustrate RM performance, we follow the evaluation approach from prior studies [53; 130], categorizing the evaluation sets by task type, as shown in Figure 3. POLAR exhibits outstanding generalization, consistently outperforming baseline RMs across most tasks. Notably, on the STEM task, POLAR-1.8B and POLAR-7B surpass the best baseline by over 24.9 and 26.2 percentage points, respectively. POLAR also accurately identifies subtle distinctions in trajectories for challenging tasks like Reasoning and general tasks such as Chat and Creative Writing, accurately predicting human preferences. The superior performance of POLAR can be attributed to its unique training paradigm. Unlike traditional RMs that rely on absolute preferences and thus perform well primarily within distribution, POLAR RMs leverage large-scale criterion-agnostic pre-training to learn nuanced differences between policies, resulting in robust out-of-distribution generalization. Notably, POLAR-1.8B achieves competitive results with just 1.8B parameters, which is only 1/15th of Skywork-Reward-27B and 1/40th of WorldPM-72B-UltraFeedback. These results highlight POLAR's efficiency and scalability.

We observed that POLAR-1.8B and POLAR-7B exhibit similar performance in preference evaluations. However, in downstream RL experiments (see Section 4.3), POLAR-7B demonstrates a notable advantage over POLAR-1.8B. This discrepancy between preference evaluations and actual RL tasks highlights an important limitation in traditional evaluation methodologies. Specifically, standard preference evaluation datasets may inadequately reflect the full spectrum of capabilities and nuanced distinctions reward models possess [130; 109; 93], underscoring the need for more comprehensive and representative evaluation frameworks in future studies.

## 4.3 Performance in RLHF Training

**Setup and Implementations** We select four open-source LLMs as policies: InternLM3-8B-Instruct [12], LLaMa-3.1-8B-Instruct [27], Qwen2.5-7B-Instruct [120], and Qwen2.5-32B-Instruct [120]. Policy optimization employs the Proximal Policy Optimization (PPO) algorithm [96]. Unlike traditional baseline RMs, which directly evaluate policy trajectories without references, POLAR requires a reference trajectory. During RL training, we assess trajectories by measuring their consistency with the provided reference, adopting the Reinforcement Fine-Tuning (RFT). Comprehensive training details are available in Appendix E.3. Following previous studies [57; 12; 121], we evaluate policy

Table 1: Reward model performance comparison in RLHF training. **Baseline** denotes the initial policy model without RLHF. We compare POLAR against SOTA reward models across 20 benchmarks within the RLHF framework. Complete results are detailed in Tables 9, 10, 11, and 12.

| Policy Model | Reward Model | General Task | Instruct Following | Coding | General Reasoning | Math | Knowledge | Average |
|---|---|---|---|---|---|---|---|---|
| **InternLM3-8B-Instruct** | Baseline | 24.07 | 62.65 | 74.40 | 64.37 | 83.11 | 60.94 | 56.49 |
| | InternLM2-Reward-7B | 28.02 | 64.45 | 78.63 | 64.84 | 79.96 | 60.43 | 57.82 |
| | Skywork-Reward-8B | 29.21 | 63.75 | 74.66 | 64.82 | 83.36 | 59.95 | 57.92 |
| | InternLM2-Reward-20B | 28.76 | 66.75 | 74.16 | 64.97 | 82.20 | 60.65 | 58.09 |
| | Skywork-Reward-27B | 30.20 | 64.95 | 74.35 | 65.18 | 83.23 | 59.91 | 58.34 |
| | WorldPM-72B-UltraFeedback | 34.89 | 67.90 | 77.13 | 65.56 | 84.29 | 61.08 | 60.49 |
| | POLAR-1.8B (Ours) | **37.50** | 72.70 | 78.24 | 66.79 | 84.33 | 64.40 | 62.60 |
| | POLAR-7B (Ours) | 37.35 | **73.25** | **79.63** | **67.89** | **85.18** | **64.46** | **63.18** |
| **Llama-3.1-8B-Instruct** | Baseline | 15.59 | 63.35 | 70.69 | 52.95 | 67.60 | 49.39 | 47.36 |
| | InternLM2-Reward-7B | 25.37 | 60.80 | 59.24 | 54.15 | 65.21 | 46.35 | 48.06 |
| | Skywork-Reward-8B | 24.80 | 61.80 | 67.53 | 53.54 | 66.23 | 49.36 | 49.22 |
| | InternLM2-Reward-20B | 26.52 | 62.85 | 58.57 | 52.41 | 64.45 | 45.09 | 47.70 |
| | Skywork-Reward-27B | 24.57 | 61.70 | 66.34 | 54.58 | 66.25 | 49.97 | 49.44 |
| | WorldPM-72B-UltraFeedback | 21.36 | 63.85 | 70.86 | 54.74 | 69.56 | 49.70 | 49.64 |
| | POLAR-1.8B (Ours) | 27.96 | 65.20 | 71.35 | 57.52 | 71.11 | 51.30 | 52.71 |
| | POLAR-7B (Ours) | **37.02** | **69.30** | **72.14** | **59.85** | **72.20** | **51.69** | **56.33** |
| **Qwen2.5-7B-Instruct** | Baseline | 26.52 | 66.05 | 79.24 | 53.83 | 83.47 | 61.98 | 54.95 |
| | InternLM2-Reward-7B | 31.99 | 64.05 | 72.80 | 56.48 | 80.35 | 55.24 | 54.95 |
| | Skywork-Reward-8B | 32.44 | 68.00 | 76.71 | 58.09 | 83.13 | 58.12 | 57.04 |
| | InternLM2-Reward-20B | 33.05 | 68.40 | 74.06 | 55.41 | 82.62 | 58.36 | 56.15 |
| | Skywork-Reward-27B | 34.28 | 69.45 | 78.21 | 57.46 | 83.58 | 59.43 | 57.84 |
| | WorldPM-72B-UltraFeedback | 35.72 | 70.55 | 77.48 | 59.48 | 83.35 | 59.48 | 58.83 |
| | POLAR-1.8B (Ours) | 35.76 | **71.35** | 80.40 | **62.52** | 84.19 | 60.39 | 60.35 |
| | POLAR-7B (Ours) | **37.70** | 71.15 | **81.15** | 61.30 | **84.70** | **62.57** | **60.90** |
| **Qwen2.5-32B-Instruct** | Baseline | 31.07 | 75.50 | 86.56 | 69.74 | 89.35 | 71.07 | 64.49 |
| | InternLM2-Reward-7B | 36.10 | 75.70 | 83.13 | 69.72 | 87.20 | 64.99 | 64.29 |
| | Skywork-Reward-8B | 36.44 | 79.60 | 84.42 | 71.07 | 89.29 | 68.77 | 66.08 |
| | InternLM2-Reward-20B | 37.98 | 74.45 | 85.76 | 69.32 | 89.35 | 66.68 | 65.25 |
| | Skywork-Reward-27B | 38.43 | 80.10 | 83.95 | 71.93 | 86.78 | 69.15 | 66.64 |
| | WorldPM-72B-UltraFeedback | 40.59 | 78.65 | 86.79 | 70.38 | 89.90 | 69.05 | 67.15 |
| | POLAR-1.8B (Ours) | 40.24 | 80.25 | 87.47 | 72.23 | 90.03 | **73.67** | 68.55 |
| | POLAR-7B (Ours) | **45.98** | **80.50** | **88.92** | **73.17** | **90.39** | 73.59 | **70.47** |

performance on 20 popular benchmarks (Table 8) using OpenCompass [18], employing the optimal settings recommended by OpenCompass for fair comparison.

**Results Comparison** Tables 1, 9, 10, 11, and 12 summarize the results from RLHF experiments using POLAR and baseline RMs. POLAR RMs consistently outperform traditional non-pre-trained RMs. For instance, the Llama-3.1 fine-tuned using POLAR-7B achieves an average improvement of 9.0% over the initial policy model and 6.7% over the policy model optimized by WorldPM-72B-UltraFeedback across all benchmarks. These results align with our findings from preference evaluations. The enhanced generalization and reduced bias in POLAR's reward signals primarily stem from its novel pre-training paradigm, which allows RMs to learn subtle distinctions between policies from extensive pre-training data rather than relying solely on labeled preference pairs. Moreover, the incorporation of reference trajectories further clarifies optimization objectives, making policy training directions more explicit and stable, thus reducing deviations and improving robustness across diverse and potentially out-of-distribution tasks. Notably, although POLAR-1.8B and POLAR-7B exhibit similar performance in preference evaluations, POLAR-7B demonstrates a significant advantage in downstream RL applications. The substantial performance gains from 1.8B to 7B parameters further illustrate the significant scaling effects achievable with the POLAR paradigm.

## 4.4 Scaling Laws in POLAR

Previous studies [27; 87] have demonstrated that scaling up neural language models consistently reduces the validation loss $L$ in a predictable manner. This reduction follows a power-law relationship relative to model parameters ($N$), training tokens ($D$), or computational resources ($C$) [50; 11]: $L = \beta \cdot X^{\alpha}$, where $X$ represents $N$, $D$, or $C$. In this formulation, $\alpha$ denotes the scaling exponent, indicating the rate at which validation loss decreases as scale $X$ increases, while $\beta$ is a normalization constant that determines the baseline level of the loss curve. Scaling laws thus facilitate the prediction of performance for larger models based on small-scale experiments, serving as an invaluable guide for efficient resource allocation and streamlined model development [95; 46; 40].

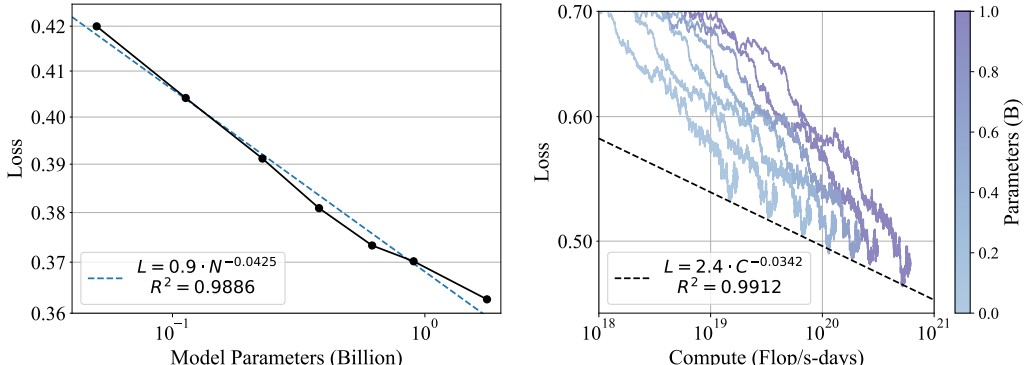

Figure 4: Scaling laws in POLAR. Validation loss vs. **(left)** model parameters $N$ and **(right)** optimal training compute $C$. Dashed lines show the power-law fit, with $R^2 = 0.9886$ (left) and $R^2 = 0.9912$ (right). Results show a predictable decrease in validation loss as model size or compute increases.

**Setup**    We explore whether POLAR also exhibits scaling laws, which would underscore its scalability and potential. Specifically, we analyze how validation loss varies with respect to model parameters ($N$) and optimal training compute ($C$), provided that sufficient data is available. To this end, we train five RMs of different sizes, ranging from 50M to 1B parameters, using up to 54B training tokens.

**Empirical Results**    We first investigate the scaling behavior of POLAR as the number of model parameters $N$ increases. As shown in the left side of Figure 4, we observe a clear power-law relationship between validation loss and model size. The fitted scaling law is given by:

$$L = 0.9 \cdot N^{-0.0425}. \tag{9}$$

The high $R^2$ value (i.e., $0.9886$) of the fit indicates an excellent match between the empirical data and the power-law trend. These findings highlight the scalability of POLAR, showing that expanding the model size consistently leads to enhanced performance.

We also examine the relationship between validation loss and optimal training compute $C$. For each model, we track the validation loss as a function of $C$ throughout training. The right panel of Figure 4 shows that validation loss follows a power-law scaling trend with respect to compute:

$$L = 2.4 \cdot C^{-0.0342}. \tag{10}$$

The high $R^2 = 0.9912$ of the fit indicates that POLAR reliably benefits from increased compute, further validating the applicability of scaling laws to our approach. This suggests that allocating more computational resources consistently yields better RM performance. Overall, these findings demonstrate that POLAR exhibits clear scaling laws similar to those observed in LLMs [41; 39]. Such predictable improvements underscore the strong potential of POLAR as a foundation for building more general reward models.

## 4.5    Ablation Study

**Impact of POLAR Pre-training**    We first investigate whether the strong performance of POLAR primarily stems from its pre-training stage. To do this, we train reward models under identical experimental settings but only utilizing the fine-tuning stage without the POLAR pre-training. Tables 2 and 13 report the performance results on preference evaluation and RLHF training, respectively. Results show these fine-tuning-only RMs achieve competitive results in preference evaluation but demonstrate a substantial performance decline in RLHF training. Consistent with prior studies [130; 109; 93], strong performance in preference evaluation does not guarantee effective reward signals for RLHF training. These observations underscore the critical role of RLHF effectiveness as an indicator of reward model quality, highlighting that POLAR's pre-training stage significantly enhances both RM performance and generalization capability.

We also trained a traditional non-pre-trained reward model without reference trajectories using the same human criteria data. The experimental results show that, across nearly all tasks, the policy

Table 2: Ablation study in RLHF training. **Baseline** denotes the initial policy model without RLHF, i.e., InternLM3-8B-Instruct [12]. **w/o PT** denotes the RM fine-tuned solely on human criteria, without any pre-training phase. **w/o PT & Ref** represents the RM trained via the traditional non-pre-trained method without reference trajectories. More detailed results are demonstrated in Table 14.

| Reward Model | | General Task | Instruct Following | Coding | General Reasoning | Math | Knowledge | Average |
|---|---|---|---|---|---|---|---|---|
| Baseline | | 24.07 | 62.65 | 74.40 | 64.37 | 83.11 | 60.94 | 56.49 |
| 1.8B | POLAR | 37.50 | 72.70 | 78.24 | 66.79 | 84.33 | 64.40 | **62.60** |
| | w/o PT | 30.37 | 68.05 | 76.96 | 65.23 | 83.30 | 63.04 | 59.45 |
| | w/o PT & Ref | 27.67 | 63.65 | 75.49 | 64.89 | 82.50 | 60.36 | 57.60 |
| 7B | POLAR | 37.35 | 73.25 | 79.63 | 67.89 | 85.18 | 64.46 | **63.18** |
| | w/o PT | 31.14 | 68.25 | 78.99 | 67.53 | 83.80 | 64.06 | 60.76 |
| | w/o PT & Ref | 31.07 | 68.30 | 76.74 | 66.16 | 83.05 | 62.06 | 59.73 |

Table 3: Ablation study on RFT vs. SFT. **RFT** denotes the reinforcement fine-tuning using POLAR. **SFT** denotes the straightforward supervised fine-tuning. Two training processes employ the same prompt-reference data. More detailed results are shown in Table 15.

| Policy Model | Method | General Task | Instruct Following | Coding | General Reasoning | Math | Knowledge | Average |
|---|---|---|---|---|---|---|---|---|
| InternLM3-8B-Instruct | Baseline | 24.07 | 62.65 | 74.40 | 64.37 | 83.11 | 60.94 | 56.49 |
| | SFT | 25.27 | 67.55 | 75.21 | 62.26 | 80.38 | 60.89 | 56.44 |
| | RFT$_{POLAR-7B}$ | 37.35 | 73.25 | 79.63 | 67.89 | 85.18 | 64.46 | **63.18** |
| Qwen2.5-7B-Instruct | Baseline | 26.52 | 66.05 | 79.24 | 53.83 | 83.47 | 61.98 | 54.95 |
| | SFT | 26.36 | 66.86 | 72.94 | 57.76 | 80.12 | 58.36 | 54.66 |
| | RFT$_{POLAR-7B}$ | 37.70 | 71.15 | 81.15 | 61.30 | 84.70 | 62.57 | **60.90** |
| Qwen2.5-32B-Instruct | Baseline | 31.07 | 75.45 | 86.56 | 69.74 | 89.35 | 71.07 | 64.49 |
| | SFT | 39.39 | 78.39 | 83.18 | 67.86 | 87.93 | 71.09 | 65.82 |
| | RFT$_{POLAR-7B}$ | 45.98 | 80.50 | 88.92 | 73.17 | 90.39 | 73.59 | **70.47** |

model fine-tuned with the RM **w/o PT** consistently outperforms the policy model fine-tuned with the RM **w/o PT & Ref**. This indicates that, even in the absence of pre-training, reference trajectories provide crucial guidance, simplifying the reward model's evaluation task and thereby yielding clearer and more stable directions for policy training.

**RFT vs. SFT** In RLHF training, POLAR RMs utilize provided references to generate reward signals for policy training, a process referred to as RFT. To assess whether performance improvements stem specifically from POLAR or merely from the reference data, we directly fine-tune policies using a straightforward SFT approach on the same reference data, as shown in Table 3. Results reveal a significant performance decline for policies trained with SFT compared to those trained with POLAR RMs using RFT. This clearly demonstrates that RL exploits training data more robustly, and POLAR RMs serve effectively as graders to provide more accurate and robust supervision signals during the training process, reliably evaluating and guiding policy optimization.

## 5 Conclusion

We propose a novel perspective on RM by reformulating it as a policy discriminator and introduce a scalable approach named Policy Discriminative Learning (POLAR). Leveraging POLAR, the RM attains robust capabilities in distinguishing among diverse policies and accurately quantifying their differences. These measured differences serve as effective reward signals to guide subsequent policy optimization. Extensive evaluations show that, despite having only small parameters, POLAR RMs significantly surpass larger SOTA baseline RMs, consistently achieving higher preference accuracy across multiple tasks. Additionally, POLAR shows substantial effectiveness in RLHF, providing reliable and generalizable reward signals even in diverse and potentially out-of-domain scenarios. Furthermore, empirical analysis confirms that POLAR exhibits clear power-law scaling behaviors, underscoring its considerable potential for future enhancements.

## Acknowledgments

The authors wish to thank the AC and anonymous reviewers for their constructive comments. This work was partially funded by Shanghai Municipal Science and Technology Major (Project 2025SHZDZX025G07), National Natural Science Foundation of China (No. 62206057, 62376061, 62476061), Shanghai Rising-Star Program (23QA1400200), and Natural Science Foundation of Shanghai (23ZR1403500).

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

# Appendix

## A    Limitations and Future Work

**Pros and Cons of Reference Trajectories**    Reference trajectories play a dual role in reward modeling. On the positive side, they significantly enhance the accuracy and reliability of the reward signal, and incorporating multiple references per prompt could further reduce reward variance, particularly in open-ended scenarios. However, reliance on reference trajectories also increases annotation costs, since generating and curating references requires substantial resources. To mitigate this limitation, we have conducted preliminary explorations suggesting that trajectories derived from other prompts with similar underlying reasoning structures may also serve as effective references. This finding indicates that the model is not strictly tied to prompt-specific references and can still assign higher rewards to correct responses even when the reference originates from a different prompt. Such a capability has the potential to reduce annotation costs while improving the flexibility and general applicability of the reward model. In future work, we plan to systematically evaluate this cross-prompt referencing strategy and investigate its integration with multiple-reference settings to further enhance robustness and performance on open-ended tasks.

**Integrating POLAR with Test-Time Scaling**    While POLAR has demonstrated effectiveness in scaling RMs for policy discrimination, it primarily focuses on pre-training paradigms. Recent advancements in test-time scaling techniques have showcased the potential of dynamically refining reasoning processes during inference via mechanisms like extended computation and self-reflection, such as OpenAI o-series reasoning models and Deepseek R1 [35; 47; 73; 48]. Such methods enable models to adaptively allocate computational resources, significantly improving decision-making quality in challenging scenarios. In future work, we aim to explore the integration of POLAR's pre-training strategies with test-time scaling techniques, investigating how such a combination can synergistically enhance RM's performance and generalization in complex tasks.

**Exploring Scaling Potential of POLAR**    Given the observed scaling law behavior, we anticipate that the current POLAR series has substantial room for further performance improvements. In future research, we plan to leverage greater computational resources to train larger-scale POLAR RMs. The data-preparing strategy employed by POLAR can be effectively scaled up to extensive pre-training datasets; however, this process inherently demands substantial policy sampling. Compared to traditional LLM data preparation, generating sufficient training data for POLAR is likely to incur considerably higher computational costs. By scaling up model size and computational resources, we aim to thoroughly investigate the limits of POLAR and release stronger, open-source models, thus facilitating continued advancements within the research community.

## B    Broader Impact

In this work, we introduce a novel method, POLAR, to pre-train the reward model and expand the potential and applicability of RL algorithms, such as RFT, paving the way for more innovative and diverse applications. The impressive performance, strong generalization, and scaling properties of our models suggest that POLAR is a promising direction for developing general and strong reward models. We do not see any negative societal impacts of this work.

## C    License For Artifacts and Data Consent

In this paper, we use the latest versions of all models and datasets provided by OpenCompass[2]. Most of the models and datasets are licensed, including the licenses for AlpacaEval and CMMLU are CC-BY-NC 4.0; the licenses for Skywork-Reward-8B and Skywork-Reward-27B can be used for academic papers; the licenses for Arena-Hard, FollowBench, FoFo, KOR-Bench, InternLM2-Reward-7B and InternLM2-Reward-20B are Apache 2.0; the licenses for WildBench, MT-Bench, MBPP and GPQA are CC-BY 4.0; the licenses for HumanEval, BBH, HellaSwag, MuSR, GSM8K, MATH-500, MMLU-Pro and MMMLU-Lite are MIT; the license for DROP is CC-BY-SA 4.0. For models and

---

[2]https://github.com/open-compass/opencompass

datasets without explicit licenses, we have actively reached out to the authors. All models and datasets used in this paper are available for academic research work.

# D  Training Details

## D.1  Pre-training

### D.1.1  Pre-training Data

We construct a synthetic pre-training dataset for POLAR by generating prompt-trajectory pairs using a diverse collection of LLMs. Specifically, we first sample prefixes from the pre-training corpus typically used for LLM training. For each piece of text from the corpus, we randomly select an initial segment as the prefix, with lengths varying randomly between 1 and 1024 tokens. Subsequently, based on each prefix, various policies autoregressively generate trajectories constrained to a maximum length of 4096 tokens. For each prefix, we randomly select two different policies from a policy pool, which contains LLMs with varying architectures and parameter scales. The first policy generates two trajectories: one serves as the reference, and the other serves as the positive trajectory. Conversely, the second policy generates a single trajectory from the same prefix, acting as a negative sample. Additionally, to enrich dataset diversity, we include a small proportion of prompts that represent human instructions, which are derived from instruction-tuning datasets. Instruction-tuned LLMs are then used to generate responses as trajectories for these instruction-based prompts.

Table 4 summarizes the composition of trajectories used for RM pre-training, totaling 3.60T tokens. Table 5 enumerates the 53 open-source pre-trained (base) LLMs and 53 open-source instruction-tuned (chat) LLMs in our policy pool for generating these trajectories. To further enhance policy diversity and ensure broader distributional coverage, we additionally incorporate 78 intermediate training checkpoints from a single pre-trained LLM——InternLM3-8B-base.

Table 4: Composition of the pre-training dataset mixture.

| Generated by | # Tokens | # Policy Models |
|---|---|---|
| Base LLMs | 3.56T | 131 |
| Chat LLMs | 0.04T | 53 |

All policies span the period from December 2023 to the present and were specifically chosen based on variations in pre-training data versions and the diversity exhibited in their generated responses.

During data construction, we encountered several issues that affected data quality. Firstly, certain policies tended to fall into repetitive loops, continuously generating identical or very similar content without meaningful progression. Instead of completely filtering out these outputs, we truncated repeated sections while preserving their loop-prone characteristics. Secondly, some generated trajectories were excessively long and lacked a clearly defined endpoint. To resolve this, we imposed a maximum output length of 4096 tokens and truncated any incomplete endings, ensuring each generated sequence was self-contained. Lastly, we observed that a low sampling temperature resulted in insufficient contrast for effective contrastive learning, while a high sampling temperature introduced biases, adversely impacting policy characterization. Based on these insights, we set the sampling temperature to 1.0, top-p to 0.9, and top-k to 50 in our trajectory sampling process.

### D.1.2  Hyperparemeter and Implementations

Our primary experiments are conducted using RMs containing 1.8 billion parameters and 7 billion parameters, denoted as POLAR-1.8B and POLAR-7B, respectively. The model architecture is detailed in Section 4.1, and we adopt the XTuner[3] framework for pre-training and fine-tuning. Rather than training from scratch, we initialize the RM from a pre-trained InternLM2.5-series model and perform one additional epoch of POLAR pre-training.

To identify optimal hyperparameters for pre-training POLAR RMs, we carried out scaling experiments designed to establish data-driven scaling laws. These scaling laws relate the optimal hyperparameters to the model size ($N$), base model pre-training data size ($D_p$), and reward model pre-training data size ($D_{rm}$). The results of these scaling experiments are illustrated in Figures 5 and 6.

---

[3]https://github.com/InternLM/xtuner

Table 5: List of Base and Chat LLMs used for generating trajectories for gathering pre-training data.We mainly utilized the Llama [33] series, Qwen [4; 106; 120] series, Yi [122] series, ChatGLM [32] series, DeepSeek[7] series, Gemma[103] series, and InternLM[104; 12] series models.

| Base LLMs | | | |
|---|---|---|---|
| Llama-3.2-1B | Llama-3.2-3B | Meta-Llama-3-70B | Meta-Llama-3-8B |
| Meta-Llama-3.1-70B | Meta-Llama-3.1-8B | Qwen-14B | Qwen-1_8B |
| Qwen-72B | Qwen-7B | Qwen1.5-0.5B | Qwen1.5-1.8B |
| Qwen1.5-14B | Qwen1.5-32B | Qwen1.5-4B | Qwen1.5-72B |
| Qwen1.5-7B | Qwen2-0.5B | Qwen2-1.5B | Qwen2-72B |
| Qwen2-7B | Qwen2.5-0.5B | Qwen2.5-1.5B | Qwen2.5-14B |
| Qwen2.5-32B | Qwen2.5-3B | Qwen2.5-72B | Qwen2.5-7B |
| Yi-1.5-34B | Yi-1.5-6B | Yi-1.5-9B | Yi-34B |
| Yi-34B-200K | Yi-6B | Yi-6B-200K | Yi-9B |
| Yi-9B-200K | chatglm3-6b-base | deepseek-llm-67b-base | deepseek-llm-7b-base |
| gemma-2b | gemma-7b | glm-4-9b | internlm-20b |
| internlm-7b | internlm2-1_8b | internlm2-20b | internlm2-7b |
| internlm2-base-20b | internlm2-base-7b | internlm2_5-1_8b | internlm2_5-20b |
| internlm2_5-7b | | | |

| Chat LLMs | | | |
|---|---|---|---|
| Llama-3.2-1B-Instruct | Llama-3.2-3B-Instruct | Meta-Llama-3-70B-Instruct | Meta-Llama-3.1-70B-Instruct |
| Meta-Llama-3.1-8B-Instruct | Qwen-14B-Chat | Qwen-1_8B-Chat | Qwen-72B-Chat |
| Qwen-7B-Chat | Qwen1.5-0.5B-Chat | Qwen1.5-1.8B-Chat | Qwen1.5-14B-Chat |
| Qwen1.5-32B-Chat | Qwen1.5-4B-Chat | Qwen1.5-72B-Chat | Qwen1.5-7B-Chat |
| Qwen2-0.5B-Instruct | Qwen2-1.5B-Instruct | Qwen2-72B-Instruct | Qwen2-7B-Instruct |
| Qwen2.5-0.5B-Instruct | Qwen2.5-1.5B-Instruct | Qwen2.5-14B-Instruct | Qwen2.5-32B-Instruct |
| Qwen2.5-3B-Instruct | Qwen2.5-72B-Instruct | Qwen2.5-7B-Instruct | Yi-1.5-34B-Chat |
| Yi-1.5-34B-Chat-16K | Yi-1.5-6B-Chat | Yi-1.5-9B-Chat | Yi-34B-Chat |
| Yi-6B-Chat | chatglm3-6b | chatglm3-6b-32k | deepseek-llm-67b-chat |
| deepseek-llm-7b-chat | gemma-2b-it | gemma-7b-it | glm-4-9b-chat |
| glm-4-9b-chat-1m | internlm2-chat-20b | internlm-chat-7b | internlm2-chat-1_8b |
| internlm2-chat-1_8b-sft | internlm2-chat-20b | internlm2-chat-20b-sft | internlm2-chat-7b |
| internlm2-chat-7b-sft | internlm2_5-1_8b-chat | internlm2_5-20b-chat | internlm2_5-7b-chat |
| internlm2_5-7b-chat-1m | | | |

In the first stage, we select base models pre-trained with various combinations of $N$ and $D_p$ and train them on reward model datasets of varying sizes ($D_{\rm rm}$). We fix the batch size at 256 and explore learning rates (LR) ranging from 5e-6 to 1e-4 for models with parameters between 100M and 924M. For each configuration, we calculate the validation loss and fit a quartic polynomial to the $\text{loss} - \log(\text{LR})$ relationship. We identify the optimal learning rate corresponding to the minimum loss for each combination of $N$, $D_p$, and $D_{\rm rm}$, resulting in the empirical formula:

$$\text{LR} = 0.0002306 \cdot N^{0.01125} \cdot D_p^{-0.66587} \cdot D_{\rm rm}^{0.33916}$$

where $N$, $D_p$, and $D_{\rm rm}$ are all expressed in millions. In the second stage, we search for the optimal batch size within the range of 128 to 1024. Learning rates are computed using the above formula and proportionally scaled based on batch size adjustments relative to the baseline of 256. Employing a similar method, we fit a quadratic polynomial to the $\text{loss} - \log(\text{batch size})$ curve and obtain:

$$\text{Batch Size} = 31.9032 \cdot N^{0.06944} \cdot D_{\rm rm}^{0.52997}$$

Finally, for POLAR-1.8B, by substituting the values $N = 1.8\text{B}$, $D_p = 2.5\text{T}$, and $D_{\rm rm} = 0.94\text{T}$, we determine the final learning rate to be 1.4e-5 and the batch size as 1940. The pre-training process is conducted on 320 NVIDIA H800 GPUs for a total duration of 57 hours.

For pre-training POLAR-7B, we set $N = 7\text{B}$, $D_p = 4.0\text{T}$, and $D_{\rm rm} = 3.6\text{T}$. Then we get the learning rate 1.67e-5 and the batch size 4343. The pre-training process is conducted on 912 NVIDIA H800 GPUs for a total duration of 175 hours.

### D.2 Supervised Fine-tuning

#### D.2.1 Supervised Fine-tuning Data

We construct a dataset comprising 150K manually labeled examples to fine-tune the pre-trained reward model. Each example includes a single prompt associated with three candidate outputs.

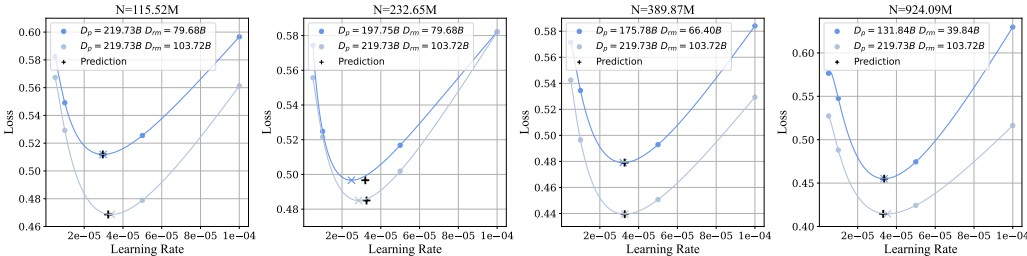

Figure 5: Scaling law of the learning rate with respect to model size and data scale in pre-training.

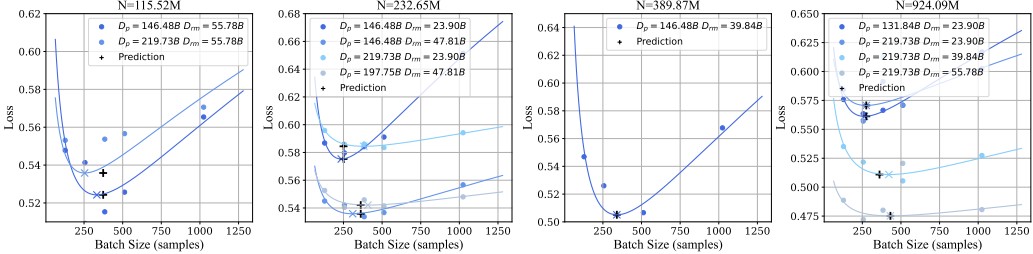

Figure 6: Scaling law of the batch size with respect to model size and data scale in pre-training.

The prompts are primarily sourced from widely used open-source preference-pair datasets such as UltraFeedback [19] and HH-RLHF [5; 28], with a small subset derived from real user queries submitted to online chat platforms.

For prompts obtained from open-source datasets that originally contain two candidate outputs, we generate a third candidate using a state-of-the-art LLM randomly selected from GPT-4o [77], OpenAI o1 [78], Deepseek-R1 [35], and Deepseek-V3 [62], to mitigate potential distribution biases. Human annotators then reorder these three outputs based on preference and adherence to instructions. For prompts sourced from actual user queries, we generate three candidate outputs utilizing LLMs selected either from the models listed in Table 5 or from the aforementioned state-of-the-art models. Human annotators subsequently evaluate and annotate relative preferences among these outputs. Consequently, each prompt is associated with three outputs ranked from best to worst. The top two outputs constitute a positive pair, while the second and third-ranked outputs form a negative pair.

All annotations were performed by company staff possessing relevant professional expertise and compensated at standard salary rates. To safeguard user privacy, we filtered all prompts in the training set to remove personally identifiable information.

### D.2.2 Hyperparemeter and Implementations

For supervised fine-tuning of POLAR models, we set the learning rate to 1e-5 for the 1.8B model and 2e-5 for the 7B model, use a batch size of 320, and train for one epoch. We experiment with multiple hyperparameter configurations and select the optimal one that avoids overoptimization. Each round of supervised fine-tuning runs on 16 NVIDIA H800 GPUs for approximately 0.5 hours.

For supervised fine-tuning of models in the ablation study, we adopt the same data and settings as POLAR, with two exceptions: The fine-tuning-only reward model (w/o PT) directly uses the InternLM2.5-series as the backbone, instead of POLAR's pre-trained models. The traditional reward model (w/o PT & Ref) is also fine-tuned on InternLM2.5-series and trained on the same preference pairs as POLAR, but without incorporating reference trajectories, following the mainstream reward model training paradigm.

### D.3 Compute and Data Comparisons

We further provide a comparison of compute and data requirements between our method and several representative baselines. Table 6 summarizes the pre-training data size, labeled data size, and compute consumption at both the pre-training and supervised fine-tuning stages. Missing values indicate that the corresponding numbers were not disclosed by the respective methods.

Table 6: Comparison of compute and data resources across different reward modeling approaches.

| Category | WorldPM -72B [111] | Skywork-Reward -8B [63] | POLAR-1.8B | POLAR-7B |
|---|---|---|---|---|
| Pre-training Data Size | - | - | ∼0.94T tokens | ∼3.6T tokens |
| Labeled Data Size | ∼15.1M samples (15,100K) | ∼40M samples (40,000K) | ∼150K samples | ∼150K samples |
| Pre-training Compute | - | - | 320 × H800 GPUs for ∼57h | 912 × H800 GPUs for ∼175h |
| SFT Compute | - | 64 × H800 GPUs | 8 × H800 GPUs for ∼0.5h | 16 × H800 GPUs for ∼0.5h |

Several observations can be drawn from this comparison. First, our method introduces a pre-training phase, which indeed requires additional computing resources. However, this phase is entirely unsupervised and thus does not require labeled data, resembling the pre-training paradigm of large language models. Second, the amount of labeled data required by our method is substantially smaller than that of traditional preference-based reward modeling approaches. Specifically, POLAR requires only 150K labeled samples for supervised fine-tuning, compared to tens of millions in other baselines, which dramatically reduces annotation costs.

# E  Details of Evaluation Setup

## E.1  Baseline Reward Models

**InternLM2-7B-Reward** and **InternLM2-20B-Reward** [12] are reward models built on InternLM2-Chat-7B-SFT and InternLM2-Chat-20B-SFT, respectively. They are trained on over 2.4 million preference samples, comprising both human-annotated and AI-generated data.

**Skywork-Reward-Gemma-2-27B** and **Skywork-Reward-Llama-3.1-8B** [63] are reward models based on the gemma-2-27b-it and Meta-Llama-3.1-8B-Instruct architectures, respectively. Both models are trained using the Skywork Reward Data Collection, which consists of 80K high-quality preference pairs curated from publicly available sources. Notably, Skywork-Reward-Gemma-2-27B achieves state-of-the-art performance in several prior studies.

**WorldPM-72B-UltraFeedback** [111] is a SOTA reward model recently released by the Qwen Group. It is initially trained on 15M preference pairs collected from public forums spanning diverse user communities, capturing a unified representation of human preferences. The model is then fine-tuned on 100K preference pairs from UltraFeedback [19], a fine-grained and diverse preference dataset.

## E.2  Details of Preference Evaluation

For the preference evaluation set constructed from real user queries submitted to online platforms (as mentioned in Section 4.2), we generate three candidate trajectories per query using either the LLMs listed in Table 5 or the state-of-the-art models mentioned earlier. Human annotators are then instructed to rank these candidate outputs based on quality. To ensure a fair evaluation, we carefully exclude any examples overlapping with the training dataset. Following the same evaluation protocol as in the RMB set, we assess the reward model's capability to correctly identify the better of the two remaining trajectories, given the highest-ranked output as a reference.

## E.3  Details of RLHF Training and Evaluation

**Training Data**  We construct a dataset containing 1.25 million prompts paired with reference trajectories to support RLHF training for policy models. The prompts are primarily drawn from widely adopted open-source instruction datasets such as UltraFeedback [19] and HH-RLHF [5; 28], supplemented by a smaller subset collected from real user queries on online chat platforms. To promote better generalization, we ensure no overlap exists between prompts used for RLHF and those utilized during SFT. For each prompt, a reference trajectory is generated by randomly selecting a

state-of-the-art LLM from GPT-4o [77], OpenAI o1 [78], Deepseek-R1 [35], and Deepseek-V3 [62]. Notably, our RLHF dataset is constructed entirely without the involvement of human annotators.

**Hyperparameter and Implementations**  For RLHF experiments, we train policy models using the PPO algorithm implemented in OpenRLHF [43], guided by both our proposed reward model and several baseline reward models. To ensure robustness, experiments are conducted across multiple random seeds, and we report the results averaged over these runs. To maintain fairness in evaluation, each baseline reward model is tested under two scoring conditions (similar to preference evaluation) when assigning rewards during PPO training: (1) standard scoring without reference trajectories, and (2) scoring with reference trajectories explicitly included in the prompt (Figure 10). For each baseline, we report the best performance observed across these two settings. Notably, we consistently find that standard scoring without references produces superior results, likely due to its closer alignment with the original training objectives of the baseline reward models.

During PPO training, we set the actor learning rate to $1e-6$, the critic learning rate to $1e-5$, the training batch size to 1024, the rollout batch size to 1024, and the number of epochs to 1. For all other hyperparameters, we generally follow OpenRLHF's recommended configurations. PPO experiments for all policy models, except Qwen2.5-32B-Instruct, are conducted using 32 NVIDIA H800 GPUs per run, each taking approximately 48 hours. For Qwen2.5-32B-Instruct, we utilize 64 NVIDIA H800 GPUs per run, with each run lasting roughly 72 hours.

# F   Additional Results

## F.1   Pre-Training Loss Curve

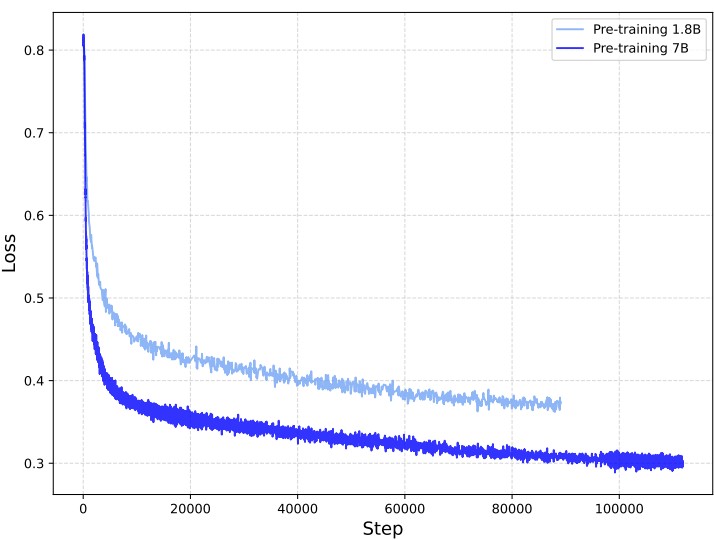

Figure 7: Pre-training loss curves for POLAR-1.8B and POLAR-7B.

Figure 7 presents the relationship between training steps and loss for POLAR-1.8B and POLAR-7B, demonstrating a steady decline in loss and indicating stable convergence during pre-training.

## F.2   Reference-free vs. Reference-included Evaluation

Table 7 reports the performance of POLAR-1.8B and two SOTA baselines (InternLM2-Reward-20B and Skywork-Reward-27B) under both reference-free and reference-included evaluation settings.

The results indicate that providing a reference does not necessarily improve the performance of traditional reward models. In many tasks, including a reference even reduces accuracy. By contrast, POLAR consistently achieves superior performance across categories.

Table 7: Performance comparison under reference-free and reference-included evaluation, and comparison between our method and similarity-based methods. INT denotes InternLM2-Reward-20B. SKY denotes Skywork-Reward-27B. EMBED denotes OpenAI's Text-Embedding-3-Large.

| Category | POLAR-1.8B | INT w/o Ref | INT w/ Ref | SKY w/o Ref | SKY w/ Ref | EMBED | Pre-trained-only POLAR-1.8B |
|---|---|---|---|---|---|---|---|
| Harmlessness | 74.2% | 66.8% | 64.8% | 75.9% | 76.1% | 66.1% | 65.7% |
| NLP Tasks | 68.5% | 70.6% | 67.8% | 71.3% | 61.9% | 62.8% | 70.6% |
| Multilingual | 80.8% | 69.2% | 64.0% | 76.9% | 60.0% | 64.0% | 73.1% |
| Chat | 75.1% | 69.5% | 62.1% | 72.3% | 67.2% | 71.1% | 67.8% |
| Brainstorming | 74.8% | 74.8% | 72.3% | 73.6% | 68.6% | 61.8% | 63.5% |
| Role Playing | 70.6% | 75.0% | 67.7% | 70.6% | 69.1% | 70.6% | 60.3% |
| STEM | 79.8% | 52.4% | 60.7% | 54.8% | 52.4% | 72.9% | 66.7% |
| Instruct Following | 73.1% | 65.4% | 69.2% | 69.2% | 73.1% | 50.0% | 65.4% |
| Closed & Open QA | 80.3% | 56.3% | 52.1% | 62.0% | 56.3% | 57.1% | 70.4% |
| Creative Writing | 85.5% | 51.3% | 60.5% | 56.6% | 60.5% | 78.1% | 64.5% |
| Coding | 65.8% | 63.2% | 54.0% | 60.5% | 68.4% | 47.1% | 59.2% |
| Reasoning | 69.2% | 53.9% | 46.2% | 53.9% | 46.2% | 77.8% | 92.3% |
| **Average** | 74.0% | 66.7% | 64.5% | 72.4% | 70.6% | 65.5% | 66.3% |

## F.3 Embedding Similarity Baselines

We further compare POLAR with an embedding-based similarity approach, where the reward value is computed as the embedding similarity between the candidate trajectory and the reference. As shown in Table 7, even when using OpenAI's strongest embedding model, POLAR significantly outperforms this baseline in preference prediction.

This highlights a fundamental difference between the two paradigms. While embedding-based methods rely on surface-level similarity between responses, POLAR explicitly measures the consistency between two policies represented by these responses. This policy-level perspective enables POLAR to provide more meaningful and robust reward signals, better aligned with RL objectives.

## F.4 Impact of POLAR Supervised Fine-tuning

To assess the necessity of the fine-tuning stage in POLAR's two-stage training framework, we evaluate the preference prediction performance of a pre-trained-only model (i.e., POLAR-1.8B) and compare it with the fully trained POLAR-1.8B. Results are shown in Table 7.

The results show that, without alignment to human judgments, the pre-trained-only model fails to achieve meaningful performance in preference prediction, despite having absorbed broad knowledge during pre-training. By contrast, fine-tuning with a small amount of labeled data enables POLAR-1.8B to align its scoring mechanism with human preferences, thereby unlocking the ability to effectively leverage its pre-trained knowledge for downstream tasks. This observation mirrors the training dynamics of LLMs, where pre-training confers broad competence, but fine-tuning is essential for aligning model behavior with human intent.

## F.5 Case Study

To showcase the effectiveness of our method, Figure 8 and Figure 9 illustrate two representative examples of how POLAR-7B assigns rewards to trajectories. Detailed explanations are provided in the respective comments and captions.

---

**Reward Scoring Example 1 for POLAR-7B**

**User Prompt:**
How many 'r's are there in the word 'strawberry'?

**Reference Response:**
There are 3 'r's in the word 'strawberry'.

**Candidate Response 1:**
There are 3 'r's in the word 'strawberry'.

*# ✔ Same as the reference response.*
*# Reward:* 0.690

**Candidate Response 2:**
There are 2 'r's in the word 'strawberry'.

*# Wrong answer.*
*# Reward:* -9.523

**Candidate Response 3:**
Let's count the 'r's in 'strawberry': 's', 't', 'r', 'a', 'w', 'b', 'e', 'r', 'r', 'y'. There are 3 'r's, so the answer is 3.

*# ✔ Correct answer with correct thoughts.*
*# Reward:* -6.476

**Candidate Response 4:**
Let's count the 'r's in 'strawberry': 's', 't', 'r', 'a', 'w', 'b', 'e', 'r', 'r', 'y'. There are 2 'r's, so the answer is 3.

*# Correct answer with wrong thoughts.*
*# Reward:* -9.289

**Candidate Response 5:**
Let's count the 'r's in 'strawberry': 's', 't', 'r', 'a', 'w', 'b', 'e', 'r', 'r', 'y'. There are 3 'r's, so the answer is 2.

*# Wrong answer with correct thoughts.*
*# Reward:* -9.562

**Candidate Response 6:**
Let's count the 'r's in 'strawberry': 's', 't', 'r', 'a', 'w', 'b', 'e', 'r', 'r', 'y'. There are 2 'r's, so the answer is 2.

*# Wrong answer with wrong thoughts.*
*# Reward:* -10.937

**Candidate Response 7:**
The word 'strawberry' contains 3 'r's.

*# ✔ Correct answer.*
*# Reward:* -1.341

**Candidate Response 8:**
The word 'strawberry' contains 2 'r's.

*# Wrong answer.*
*# Reward:* -9.929

---

Figure 8: An example of reward model scoring applied to the popular "strawberry" test. Despite variations in response phrasing, our reward model accurately identifies the correctness of the final answer and distinguishes the validity of the reasoning process.

## Reward Scoring Example 2 for POLAR-7B

**User Prompt:**
Summarize the first book of Frank Herbert's Dune in one witty short sentence.

**Reference Response:**
Royal teen discovers that life's a beach—minus the ocean, plus spice, giant sandworms and deadly politics.

**Candidate Response 1:**
Royal teen discovers that life's a beach—minus the ocean, plus spice, giant sandworms and deadly politics.

*# ✔ Same as the reference response.*
*# Reward:* 0.467

**Candidate Response 2:**
Royal teen discovers that life's a beach—minus the ocean, plus magic, dark wizards and deadly politics.

*# Closely resembles the reference response but includes factual errors.*
*# Reward:* -7.711

**Candidate Response 3:**
Young noble's move to desert planet turns into galactic Game of Thrones with fewer dragons, more worms.

*# ✔ A distinct yet concise and witty summary that draws analogies from other dramas—markedly different from the reference response.*
*# Reward:* -6.918

**Candidate Response 4:**
A noble family's fall sparks a young heir's rise as a leader on a harsh desert planet governed by prophecy and survival.

*# A concise summary, but lacking wit—fails to meet the requirement.*
*# Reward:* -8.648

**Candidate Response 5:**
Paul Atreides loses his father, gains prophetic powers, learns to ride a sandworm, leads a holy war, and discovers that being the chosen one comes with a lot of blood, sand, and questionable decisions.

*# A witty summary, but overly long—fails to meet the requirement.*
*# Reward:* -8.429

**Candidate Response 6:**
Boy gets planet, becomes god, loses soul — family drama ensues across galaxies.

*# A concise and witty summary that draws from multiple Dune books rather than just the first—fails to follow the instruction.*
*# Reward:* -10.359

Figure 9: An example of reward model scoring applied to a summarization task. For open-ended questions, our model accurately evaluates the alignment of the core content of responses with the prompt and reference answer, considering multiple evaluation criteria rather than merely relying on semantic similarity to the reference response.

## F.6  Additional Results of POLAR Evaluation

Figure 10 illustrates the prompt format used by baseline reward models to score trajectories when a reference trajectory is provided, as mentioned in Section 4.2.

Table 8 shows the details of our employed benchmarks for policy evaluation. Table 9, 10, 11, and 12 show the whole results of four policies in RLHF. Table 13, 14 and 15 present additional ablation results mentioned in Section 4.5.

---

**Prompt including reference for reward model baselines**

**System:**
You are a helpful AI assistant, designed to provide useful answers to customers. The user will provide you with a question and a reference answer. Please refer to this desired output and provide your own unique response.
**User:**
You will be provided with the following information:
[The Start of conversation history] {dialog history} [The end of conversation History]
[The start of question] {question} [The end of question]
[The start of desired output] {reference} [The end of desired output]
Please think carefully about this question and the desired output, and then provide your response to the question.
**Assistant:**
{response}

---

Figure 10: The prompt format used by baseline reward models to score trajectories with a provided reference, as mentioned in Section 4.2. Specifically, we construct a dialogue scenario in which the assistant generates a response to the prompt with access to the reference, and the baseline reward models then assign scores to the response. To ensure a fair comparison with our POLAR RMs, we report the best performance of each baseline across both the reference-included and traditional (no-reference) settings.

Table 8: Benchmarks we used in policy evaluation.

| Benchmark | Category | Description |
|-----------|----------|-------------|
| **AlignBench**[67] | General Task | A comprehensive Chinese benchmark for LLM alignment evaluation. |
| **AlpacaEval**[59] | General Task | An automated benchmark for evaluating instruction-following ability. |
| **Arena-Hard**[58] | General Task | A dynamic benchmark on challenging, real-world tasks. |
| **WildBench**[61] | General Task | An automated benchmark on real-world user tasks. |
| **MT-Bench**[128] | General Task | A benchmark on multi-turn conversational tasks. |
| **FollowBench**[49] | Instruction Following | A benchmark for evaluating LLMs' ability to follow instructions with multi-level, fine-grained constraints. |
| **FoFo**[118] | Instruction Following | A benchmark for evaluating LLMs' ability to follow complex, domain-specific formats and formatting rules. |
| **HumanEval**[14] | Coding | A benchmark for code generation tasks measuring functional correctness for synthesizing programs from docstrings. |
| **MBPP**[3] | Coding | A code generation benchmark including 1,000 entry-level Python programming problems. |
| **DROP**[26] | General Reasoning | A reading comprehension benchmark requiring discrete reasoning over paragraphs. |
| **BBH**[102] | General Reasoning | A diverse benchmark comprising 23 challenging BIG-Bench tasks. |
| **GPQA**[94] | General Reasoning | A graduate-level Google-Proof Q&A benchmark covering the fields of biology, physics, and chemistry. |
| **HellaSwag**[125] | General Reasoning | A challenging dataset for evaluating commonsense NLI. |
| **MuSR**[99] | General Reasoning | A dataset on multistep soft reasoning tasks specified in a natural language narrative. |
| **KOR-Bench**[71] | General Reasoning | A challenging benchmark comprising five tasks that introduce new rules independent of prior knowledge. |
| **GSM8K**[17] | Math | A dataset of 8.5K high quality linguistically diverse grade school math word problems . |
| **MATH-500**[60] | Math | A challenging dataset of 500 high school math competition problems. |
| **CMMLU**[55] | Knowledge | A comprehensive Chinese benchmark for assessing knowledge and reasoning abilities in Chinese contexts. |
| **MMLU-Pro**[116] | Knowledge | An enhanced version benchmark containing over 12,000 academic exam and textbook questions across 14 domains. |
| **MMMLU-Lite**[98] | Knowledge | A massive multilingual multitask language understanding dataset spanning 15 languages. |

Table 9: Reward model comparison in RLHF training. The policy model is initialized from InternLM3-8B-Instruct [12] (i.e., baseline). Sky8B and Sky27B denote Skywork-Reward-8B and Skywork-Reward-8B [63], respectively. Int7B and Int20B denote InternLM2-Reward-7B and InternLM2-Reward-20B [12], respectively. WPM72B denotes WorldPM-72B-UltraFeedback [111].

| Type | Dataset | Policy model: InternLM3-8B-Instruct | | | | | | | |
| --- | --- | --- | --- | --- | --- | --- | --- | --- | --- |
| | | Baseline | Int7B | Sky8B | Int20B | Sky27B | WPM72B | POLAR-1.8B | POLAR-7B |
| General Task | AlignBench | 6.13 | 6.24 | 6.09 | 6.37 | 6.27 | 6.43 | 6.54 | 6.53 |
| | AlpacaEval | 51.80 | 57.02 | 61.37 | 58.39 | 63.48 | 70.93 | 68.57 | 68.70 |
| | Arena-Hard | 40.79 | 45.96 | 45.01 | 47.33 | 45.56 | 53.69 | 63.47 | 63.34 |
| | WildBench | 13.67 | 22.64 | 25.30 | 23.42 | 27.40 | 35.00 | 40.34 | 39.75 |
| | MT-Bench | 7.97 | 8.24 | 8.27 | 8.29 | 8.27 | 8.40 | 8.57 | 8.43 |
| Instruct Following | FollowBench | 82.20 | 83.20 | 83.60 | 85.30 | 84.80 | 86.20 | 91.10 | 90.40 |
| | FoFo | 43.10 | 45.70 | 43.90 | 48.20 | 45.10 | 49.60 | 54.30 | 56.10 |
| Coding | HumanEval | 81.10 | 82.93 | 78.88 | 78.66 | 78.66 | 81.10 | 82.93 | 84.15 |
| | MBPP | 67.70 | 74.32 | 70.43 | 69.65 | 70.04 | 73.15 | 73.54 | 75.10 |
| General Reasoning | DROP | 82.82 | 82.92 | 82.70 | 83.13 | 82.88 | 83.34 | 86.64 | 85.63 |
| | BBH | 77.56 | 75.12 | 76.99 | 76.43 | 77.27 | 77.97 | 77.06 | 77.70 |
| | GPQA | 32.83 | 36.36 | 35.35 | 36.87 | 37.37 | 37.88 | 37.37 | 41.41 |
| | HellaSwag | 88.88 | 87.15 | 87.65 | 87.50 | 87.60 | 87.06 | 88.78 | 88.87 |
| | MuSR | 52.28 | 51.48 | 51.59 | 49.64 | 51.23 | 50.41 | 57.07 | 58.10 |
| | KOR-Bench | 51.84 | 56.00 | 54.64 | 56.24 | 54.72 | 56.72 | 53.84 | 55.60 |
| Math | GSM8K | 90.22 | 89.31 | 89.92 | 89.99 | 90.45 | 90.98 | 91.05 | 91.96 |
| | MATH-500 | 76.00 | 70.60 | 76.80 | 74.40 | 76.00 | 77.60 | 77.60 | 78.40 |
| Knowledge | CMMLU | 81.70 | 82.26 | 81.90 | 82.42 | 82.18 | 82.75 | 84.79 | 84.39 |
| | MMLU-Pro | 57.49 | 58.05 | 57.94 | 57.99 | 57.96 | 58.20 | 60.00 | 59.98 |
| | MMMLU-Lite | 43.64 | 40.97 | 40.02 | 41.54 | 39.59 | 42.30 | 48.40 | 49.00 |
| Average | | 56.49 | 57.82 | 57.92 | 58.09 | 58.34 | 60.49 | 62.60 | **63.18** |

Table 10: Reward model comparison in RLHF training. The policy model is initialized from Llama-3.1-8B-Instruct [27] (i.e., baseline). Sky8B and Sky27B denote Skywork-Reward-8B and Skywork-Reward-8B [63], respectively. Int7B and Int20B denote InternLM2-Reward-7B and InternLM2-Reward-20B [12], respectively. WPM72B denotes WorldPM-72B-UltraFeedback [111].

| Type | Dataset | Policy model: Llama-3.1-8B-Instruct | | | | | | | |
| --- | --- | --- | --- | --- | --- | --- | --- | --- | --- |
| | | Baseline | Int7B | Sky8B | Int20B | Sky27B | WPM72B | POLAR-1.8B | POLAR-7B |
| General Task | AlignBench | 4.66 | 4.89 | 5.07 | 4.90 | 5.18 | 4.94 | 4.89 | 5.25 |
| | AlpacaEval | 23.73 | 48.57 | 50.01 | 51.68 | 50.56 | 41.16 | 50.19 | 72.30 |
| | Arena-Hard | 42.31 | 45.26 | 39.84 | 37.27 | 39.22 | 38.78 | 52.60 | 61.95 |
| | WildBench | -0.89 | 19.88 | 20.81 | 30.70 | 19.54 | 13.66 | 23.69 | 37.12 |
| | MT-Bench | 8.15 | 8.24 | 8.27 | 8.03 | 8.33 | 8.28 | 8.42 | 8.46 |
| Instruct Following | FollowBench | 89.70 | 91.00 | 91.20 | 90.90 | 91.20 | 90.30 | 92.10 | 92.90 |
| | FoFo | 37.00 | 30.60 | 32.40 | 34.80 | 32.20 | 37.40 | 38.30 | 45.70 |
| Coding | HumanEval | 71.34 | 54.27 | 64.63 | 59.15 | 63.41 | 70.12 | 70.72 | 70.73 |
| | MBPP | 70.04 | 64.20 | 70.43 | 57.98 | 69.26 | 71.60 | 71.98 | 73.54 |
| General Reasoning | DROP | 79.46 | 79.33 | 81.42 | 74.43 | 82.53 | 81.60 | 84.48 | 84.14 |
| | BBH | 54.26 | 49.37 | 55.57 | 58.25 | 61.79 | 60.50 | 64.34 | 70.02 |
| | GPQA | 29.80 | 32.32 | 31.88 | 31.31 | 27.78 | 28.28 | 32.83 | 39.90 |
| | HellaSwag | 46.90 | 63.56 | 53.06 | 52.60 | 51.03 | 53.99 | 60.92 | 67.84 |
| | MuSR | 60.73 | 54.22 | 51.15 | 53.76 | 56.18 | 56.86 | 55.70 | 50.73 |
| | KOR-Bench | 46.56 | 46.08 | 48.16 | 44.08 | 48.16 | 47.20 | 46.86 | 46.48 |
| Math | GSM8K | 83.40 | 83.02 | 85.06 | 83.09 | 82.49 | 85.52 | 85.82 | 88.40 |
| | MATH-500 | 51.80 | 47.40 | 47.40 | 45.80 | 50.00 | 53.60 | 56.40 | 56.00 |
| Knowledge | CMMLU | 54.78 | 52.81 | 54.94 | 52.86 | 55.63 | 54.96 | 55.29 | 53.10 |
| | MMLU-Pro | 49.57 | 48.11 | 50.14 | 47.53 | 49.84 | 50.68 | 51.57 | 52.66 |
| | MMMLU-Lite | 43.83 | 38.12 | 43.01 | 34.87 | 44.43 | 43.45 | 47.04 | 49.32 |
| Average | | 47.36 | 48.06 | 49.22 | 47.70 | 49.44 | 49.64 | 52.71 | **56.33** |

Table 11: Reward model comparison in RLHF training. The policy model is initialized from Qwen2.5-7B-Instruct [120] (i.e., baseline). Sky8B and Sky27B denote Skywork-Reward-8B and Skywork-Reward-8B [63], respectively. Int7B and Int20B denote InternLM2-Reward-7B and InternLM2-Reward-20B [12], respectively. WPM72B denotes WorldPM-72B-UltraFeedback [111].

| Type | Dataset | Policy model: Qwen2.5-7B-Instruct | | | | | | | |
|------|---------|----------|-------|-------|--------|--------|--------|------------|---------|
| | | Baseline | Int7B | Sky8B | Int20B | Sky27B | WPM72B | POLAR-1.8B | POLAR-7B |
| General Task | AlignBench | 6.18 | 6.22 | 6.27 | 6.25 | 6.30 | 6.48 | 6.42 | 6.49 |
| | AlpacaEval | 37.02 | 58.39 | 58.63 | 62.81 | 61.61 | 62.18 | 60.13 | 61.86 |
| | Arena-Hard | 56.73 | 51.88 | 54.10 | 53.81 | 59.64 | 61.52 | 67.11 | 71.38 |
| | WildBench | 24.30 | 35.10 | 34.78 | 33.99 | 35.40 | 39.99 | 36.63 | 40.30 |
| | MT-Bench | 8.38 | 8.37 | 8.40 | 8.40 | 8.43 | 8.43 | 8.51 | 8.49 |
| Instruct Following | FollowBench | 88.00 | 89.00 | 89.20 | 89.40 | 89.90 | 90.70 | 90.50 | 90.30 |
| | FoFo | 44.10 | 39.10 | 46.80 | 47.40 | 49.00 | 50.40 | 52.20 | 52.00 |
| Coding | HumanEval | 84.15 | 78.66 | 79.88 | 76.83 | 81.71 | 81.41 | 84.15 | 87.20 |
| | MBPP | 74.32 | 66.93 | 73.54 | 71.28 | 74.71 | 73.54 | 76.65 | 75.10 |
| General Reasoning | DROP | 78.35 | 78.09 | 81.10 | 78.44 | 80.25 | 80.79 | 82.67 | 83.22 |
| | BBH | 56.92 | 62.73 | 66.83 | 65.15 | 67.50 | 66.50 | 68.67 | 67.14 |
| | GPQA | 32.32 | 33.33 | 33.84 | 33.36 | 34.34 | 38.64 | 38.38 | 37.37 |
| | HellaSwag | 67.47 | 74.63 | 74.73 | 65.48 | 71.27 | 76.24 | 84.91 | 81.25 |
| | MuSR | 46.53 | 41.79 | 44.11 | 40.84 | 45.31 | 46.22 | 51.04 | 50.64 |
| | KOR-Bench | 41.36 | 48.32 | 47.92 | 49.20 | 46.08 | 48.48 | 49.44 | 48.16 |
| Math | GSM8K | 91.13 | 88.70 | 90.45 | 89.84 | 91.05 | 91.40 | 91.58 | 92.19 |
| | MATH-500 | 75.80 | 72.00 | 75.80 | 75.40 | 76.10 | 75.30 | 76.80 | 77.20 |
| Knowledge | CMMLU | 77.56 | 73.80 | 75.53 | 74.97 | 76.94 | 76.76 | 78.07 | 77.92 |
| | MMLU-Pro | 55.33 | 52.30 | 55.54 | 54.89 | 55.34 | 55.48 | 55.97 | 56.65 |
| | MMMLU-Lite | 53.04 | 39.61 | 43.28 | 45.22 | 46.01 | 46.19 | 47.13 | 53.15 |
| Average | | 54.95 | 54.95 | 57.04 | 56.15 | 57.84 | 58.83 | 60.35 | **60.90** |

Table 12: Reward model comparison in RLHF training. The policy model is initialized from Qwen2.5-32B-Instruct [120] (i.e., baseline). Sky8B and Sky27B denote Skywork-Reward-8B and Skywork-Reward-8B [63], respectively. Int7B and Int20B denote InternLM2-Reward-7B and InternLM2-Reward-20B [12], respectively. WPM72B denotes WorldPM-72B-UltraFeedback [111].

| Type | Dataset | Policy model: Qwen2.5-32B-Instruct | | | | | | | |
|------|---------|----------|-------|-------|--------|--------|--------|------------|---------|
| | | Baseline | Int7B | Sky8B | Int20B | Sky27B | WPM72B | POLAR-1.8B | POLAR-7B |
| General Task | AlignBench | 6.76 | 6.81 | 6.90 | 6.98 | 6.86 | 7.05 | 7.02 | 7.12 |
| | AlpacaEval | 37.27 | 56.19 | 58.42 | 60.75 | 60.71 | 65.66 | 61.92 | 78.88 |
| | Arena-Hard | 77.75 | 70.98 | 72.03 | 76.27 | 78.59 | 81.34 | 84.20 | 89.52 |
| | WildBench | 25.09 | 38.03 | 36.33 | 37.39 | 37.49 | 40.34 | 39.50 | 45.82 |
| | MT-Bench | 8.47 | 8.49 | 8.54 | 8.52 | 8.51 | 8.54 | 8.54 | 8.57 |
| Instruct Following | FollowBench | 91.30 | 94.10 | 94.00 | 94.00 | 94.40 | 95.00 | 94.50 | 94.20 |
| | FoFo | 59.70 | 57.30 | 65.20 | 54.90 | 65.80 | 62.30 | 66.00 | 66.80 |
| Coding | HumanEval | 90.24 | 84.15 | 90.24 | 89.02 | 90.85 | 90.00 | 90.89 | 91.46 |
| | MBPP | 82.88 | 82.10 | 78.60 | 82.49 | 77.04 | 83.58 | 84.05 | 86.38 |
| General Reasoning | DROP | 89.24 | 87.68 | 89.77 | 89.24 | 89.67 | 89.50 | 90.52 | 90.31 |
| | BBH | 79.29 | 79.34 | 82.50 | 82.02 | 83.12 | 81.43 | 81.51 | 82.49 |
| | GPQA | 43.43 | 43.94 | 46.46 | 44.44 | 47.98 | 47.47 | 47.47 | 52.02 |
| | HellaSwag | 82.40 | 83.92 | 83.62 | 76.78 | 85.77 | 80.24 | 87.98 | 87.99 |
| | MuSR | 66.70 | 65.35 | 64.17 | 64.56 | 65.19 | 64.87 | 67.64 | 67.49 |
| | KOR-Bench | 57.36 | 58.08 | 59.92 | 58.88 | 59.84 | 58.74 | 58.24 | 58.72 |
| Math | GSM8K | 95.30 | 95.00 | 95.38 | 95.45 | 91.96 | 95.56 | 96.06 | 95.98 |
| | MATH-500 | 83.40 | 79.40 | 83.20 | 83.24 | 81.60 | 84.24 | 84.00 | 84.80 |
| Knowledge | CMMLU | 84.41 | 81.70 | 82.64 | 82.86 | 83.06 | 83.73 | 86.02 | 85.52 |
| | MMLU-Pro | 68.38 | 67.56 | 69.88 | 69.82 | 70.23 | 69.15 | 69.94 | 70.42 |
| | MMMLU-Lite | 60.43 | 45.71 | 53.79 | 47.36 | 54.16 | 54.26 | 65.06 | 64.83 |
| Average | | 64.49 | 64.29 | 66.08 | 65.25 | 66.64 | 67.15 | 68.55 | **70.47** |

Table 13: Ablation study of POLAR on preference evaluation. **w/o PT** refers to the reward model that is fine-tuned solely with human criteria, without our pre-training phase. **w/o PT & Ref** denotes the reward model trained via the traditional non-pre-trained method without reference trajectories. Since strong performance in preference evaluation does not guarantee effective reward signals, we conduct an extra ablation study on RLHF training in Table 2 and Table 14, which provides a stronger validation.

| Tasks | POLAR-1.8B | | | POLAR-7B | | |
|---|---|---|---|---|---|---|
| | Origin | w/o PT | w/o PT & Ref | Origin | w/o PT | w/o PT & Ref |
| **Harmlessness** | 74.2 | 73.9 | 63.8 | 77.7 | 78.4 | 74.0 |
| **NLP Tasks** | 68.5 | 69.9 | 65.4 | 71.3 | 71.3 | 70.9 |
| **Multilingual** | 80.8 | 84.6 | 56.0 | 80.0 | 84.0 | 72.0 |
| **Chat** | 75.1 | 70.6 | 65.5 | 79.1 | 71.8 | 63.8 |
| **Brainstorming** | 74.8 | 72.3 | 67.9 | 72.3 | 79.9 | 76.1 |
| **Role Playing** | 70.6 | 60.3 | 70.6 | 67.7 | 70.6 | 63.2 |
| **STEM** | 79.8 | 75.0 | 59.5 | 81.0 | 77.4 | 54.8 |
| **Instruct Following** | 73.1 | 76.9 | 50.0 | 65.4 | 61.5 | 61.5 |
| **Closed & Open QA** | 80.3 | 73.2 | 54.9 | 80.3 | 76.1 | 64.8 |
| **Creative Writing** | 85.5 | 81.6 | 57.9 | 85.5 | 86.8 | 65.8 |
| **Coding** | 65.8 | 57.9 | 54.0 | 68.4 | 64.5 | 59.2 |
| **Reasoning** | 69.2 | 69.1 | 46.2 | 69.2 | 76.9 | 46.2 |
| **Average** | 74.0 | 72.6 | 63.3 | 76.3 | 76.5 | 70.8 |

Table 14: Ablation study of POLAR in RLHF training. **w/o PT** denotes the reward model fine-tuned solely on human criteria, without any pre-training phase. **w/o PT & Ref** represents the reward model trained via the traditional non-pre-trained method without reference trajectories.

| Type | Dataset | Policy Model: InternLM3-8B-Instruct | | | | | | |
|---|---|---|---|---|---|---|---|---|
| | | Baseline | Reward Model Size: 1.8B | | | Reward Model Size: 7B | | |
| | | | POLAR | w/o PT | w/o PT & Ref | POLAR | w/o PT | w/o PT & Ref |
| General Task | AlignBench | 6.13 | 6.54 | 6.39 | 6.19 | 6.53 | 6.39 | 6.44 |
| | AlpacaEval | 51.80 | 68.57 | 53.54 | 58.63 | 68.70 | 57.89 | 53.42 |
| | Arena-Hard | 40.79 | 63.47 | 49.84 | 42.71 | 63.34 | 54.99 | 51.68 |
| | WildBench | 13.67 | 40.34 | 33.79 | 22.64 | 39.75 | 28.06 | 35.49 |
| | MT-Bench | 7.97 | 8.57 | 8.28 | 8.17 | 8.43 | 8.35 | 8.33 |
| Instruct Following | FollowBench | 82.20 | 91.10 | 84.30 | 81.80 | 90.40 | 89.30 | 84.00 |
| | FoFo | 43.10 | 54.30 | 51.80 | 45.50 | 56.10 | 47.20 | 52.60 |
| Coding | HumanEval | 81.10 | 82.93 | 82.32 | 81.71 | 84.15 | 81.71 | 81.10 |
| | MBPP | 67.70 | 73.54 | 71.60 | 69.26 | 75.10 | 76.26 | 72.37 |
| General Reasoning | DROP | 82.82 | 86.64 | 84.22 | 83.31 | 85.63 | 86.86 | 83.89 |
| | BBH | 77.56 | 77.06 | 77.76 | 77.40 | 77.70 | 76.41 | 77.74 |
| | GPQA | 32.83 | 37.37 | 35.86 | 36.87 | 41.41 | 39.90 | 38.38 |
| | HellaSwag | 88.88 | 88.78 | 88.46 | 88.04 | 88.87 | 89.01 | 87.21 |
| | MuSR | 52.28 | 57.07 | 54.53 | 49.90 | 58.10 | 57.71 | 53.32 |
| | KOR-Bench | 51.84 | 53.84 | 50.56 | 53.84 | 55.60 | 55.28 | 56.40 |
| Math | GSM8K | 90.22 | 91.05 | 89.99 | 90.30 | 91.96 | 90.60 | 90.90 |
| | MATH-500 | 76.00 | 77.60 | 76.60 | 74.70 | 78.40 | 77.00 | 75.20 |
| Knowledge | CMMLU | 81.70 | 84.79 | 82.70 | 81.83 | 84.39 | 83.98 | 82.60 |
| | MMLU-Pro | 57.49 | 60.00 | 58.28 | 57.34 | 59.98 | 58.92 | 58.13 |
| | MMMLU-Lite | 43.64 | 48.40 | 48.15 | 41.92 | 49.00 | 49.28 | 45.46 |
| **Average** | | 56.49 | **62.60** | 59.45 | 57.60 | **63.18** | 60.76 | 59.73 |

Table 15: Ablation study of POLAR on RFT vs. SFT. **RFT** denotes the reinforcement fine-tuning using our proposed reward models. **SFT** denotes the straightforward supervised fine-tuning. Two training processes employ the same prompt-reference data. Results indicate that RFT with a reward model is significantly more effective than SFT. The substantial improvement in policy model performance cannot be attributed solely to the training data (i.e., prompt-reference data) used in RFT.

| Policy Model | | InternLM3-8B-Instruct | | | Qwen2.5-7B-Instruct | | | Qwen2.5-32B-Instruct | | |
|---|---|---|---|---|---|---|---|---|---|---|
| **Dataset** | | **Baseline** | **RFT$_{POLAR-7B}$** | **SFT** | **Baseline** | **RFT$_{POLAR-7B}$** | **SFT** | **Baseline** | **RFT$_{POLAR-7B}$** | **SFT** |
| General Task | **AlignBench** | 6.13 | 6.53 | 5.84 | 6.18 | 6.49 | 5.62 | 6.76 | 7.12 | 6.52 |
| | **AlpacaEval** | 51.80 | 68.70 | 57.39 | 37.02 | 61.86 | 56.40 | 37.27 | 78.88 | 75.53 |
| | **Arena-Hard** | 40.79 | 63.34 | 49.74 | 56.73 | 71.38 | 59.43 | 77.75 | 89.52 | 80.17 |
| | **WildBench** | 13.67 | 39.75 | 5.77 | 24.30 | 40.30 | 2.77 | 25.09 | 45.82 | 26.67 |
| | **MT-Bench** | 7.97 | 8.43 | 7.61 | 8.38 | 8.49 | 7.60 | 8.47 | 8.57 | 8.05 |
| Instruct Following | **FollowBench** | 82.20 | 90.40 | 83.60 | 88.00 | 90.30 | 88.60 | 91.30 | 94.20 | 92.30 |
| | **FoFo** | 43.10 | 56.10 | 51.49 | 44.10 | 52.00 | 45.12 | 59.60 | 66.80 | 64.47 |
| Coding | **HumanEval** | 81.10 | 84.15 | 78.05 | 84.15 | 87.20 | 76.22 | 90.24 | 91.46 | 86.59 |
| | **MBPP** | 67.70 | 75.10 | 72.37 | 74.32 | 75.10 | 69.65 | 82.88 | 86.38 | 79.77 |
| General Reasoning | **DROP** | 82.82 | 85.63 | 87.12 | 78.35 | 83.22 | 84.71 | 89.24 | 90.31 | 90.54 |
| | **BBH** | 77.56 | 77.70 | 68.22 | 56.92 | 67.14 | 59.76 | 79.29 | 82.49 | 74.99 |
| | **GPQA** | 32.83 | 41.41 | 33.33 | 32.32 | 37.37 | 27.78 | 43.43 | 52.02 | 42.93 |
| | **HellaSwag** | 88.88 | 88.87 | 88.96 | 67.47 | 81.25 | 84.16 | 82.40 | 87.99 | 90.73 |
| | **MuSR** | 52.28 | 58.10 | 53.45 | 46.53 | 50.64 | 46.01 | 66.70 | 67.49 | 62.87 |
| | **KOR-Bench** | 51.84 | 55.60 | 42.48 | 41.36 | 48.16 | 44.16 | 57.36 | 58.72 | 45.12 |
| Math | **GSM8K** | 90.22 | 91.96 | 89.76 | 91.13 | 92.19 | 88.63 | 95.30 | 95.98 | 95.45 |
| | **MATH-500** | 76.00 | 78.40 | 71.00 | 75.80 | 77.20 | 71.60 | 83.40 | 84.80 | 80.40 |
| Knowledge | **CMMLU** | 81.70 | 84.39 | 82.03 | 77.56 | 77.92 | 74.23 | 84.41 | 85.52 | 83.62 |
| | **MMLU-Pro** | 57.49 | 59.98 | 54.74 | 55.33 | 56.65 | 52.42 | 68.38 | 70.42 | 66.37 |
| | **MMLU-Lite** | 43.64 | 49.00 | 45.90 | 53.04 | 53.15 | 48.42 | 60.43 | 64.83 | 63.28 |
| Average | | 56.49 | **63.18** | 56.44 | 54.95 | **60.90** | 54.66 | 64.49 | **70.47** | 65.82 |

