# OpenReview forum: "Pre-Trained Policy Discriminators are General Reward Models"
_NeurIPS.cc/2025/Conference — NeurIPS 2025 poster_

### Official Review · Reviewer_wgBV · 2025-07-01

**Clarity:** 2
**Significance:** 3
**Originality:** 3
**Rating:** 4
**Confidence:** 3

**Summary:**

This paper introduces an alternative perspective on training reward models by framing the reward model as a policy discriminator—quantifying the differences between candidate policies and a given target policy. The proposed approach, called POEM, consists of two stages: unsupervised pretraining followed by supervised fine-tuning. Experimental results demonstrate that POEM outperforms existing reward modeling methods, achieving higher preference accuracy and improved performance in RLHF settings. Additionally, scaling law experiments suggest that increasing model size and compute resources further enhances POEM’s effectiveness.

**Questions:**

1. In line 123, should \tau be just a trajectory and not a set of trajectories?
2. Equations 1 and 2 need citations.
3. In line 140, I disagree with the comment that says “there always exists a complementary criterion”. I think it is definitely generally true, but not always.
4. In Eq. 6, what is \rho? I think it has to be a sigmoid function, but please do not assume the reader knows all notations.

**Ethical Concerns:**

["NO or VERY MINOR ethics concerns only"]

**Final Justification:**

The proposed methods bring an interesting viewpoint for reward modeling, and I believe it is of interest to the community. The rebuttal was responsive to all my major points. While the work has some limitations, as other reviewers also pointed out, like reliance on the reference trajectories and the potential sensitivity of the performance on the selected policies, the authors have acknowledged and discussed these limitations. Therefore, I lean towards an accepting score.

**Limitations:**

The authors have discussed some limitations, which is nice, but they might be missing some, as I explained previously.

**Paper Formatting Concerns:**

No concerns

**Quality:**

2

**Strengths And Weaknesses:**

On the positive side, this work takes a step towards better reward modeling approaches, which is of great interest. The alternative perspective of reward modeling as policy discrimination is interesting and likely to be of interest to the community. As shown in the experiments, this approach empirically leads to improved preference learning and RLHF performance. Also, the potential for further gains through larger models and increased compute is promising.

On the constructive side, I found that the paper lacks sufficient contextualization within the existing body of work, which makes it difficult to fully appreciate its contributions. For example, the authors argue that previous work using labeled preference pairs to train reward models is limited by the difficulty of obtaining large volumes of high-quality human annotations. However, POEM also relies on human rankings during the fine-tuning stage. Similarly, while the paper critiques the Bradley-Terry framework for assuming absolute human preferences—which may lead to misleading outcomes—POEM still employs a similar concept during fine-tuning, now requiring humans to compare three samples instead of two. These apparent inconsistencies appear throughout the paper. It is fine to have these limitations, given that the POEM empirical results seem to be better, but I would like to see a proper contextualization of the work within the existing body of literature and a proper discussion of the limitations. This would make the reader appreciate the work better. Second, given the above limitations, I wonder if the success of POEM originates from the alternative refomulation of the loss (using policy discrimination) alone? It’s possible that I may be overlooking something, as the authors have likely considered this aspect carefully, so I would appreciate the authors' thoughts on the matter.

Another aspect that would benefit from further discussion is the training of POEM reward models using Equation 6 with trajectories drawn from different policies. My assumption is that POEM’s performance is likely sensitive to the selection of these policies, potentially in a significant way. However, the current version of the paper does not include any analysis in this regard. An analysis of how the choice of policy set influences performance would make the work stronger.

Further, it would be nice to justify the two-stage training requirement for POEM. Specifically, how far can one go using only the pretraining stage? While the paper includes an ablation study highlighting the impact of pretraining, it lacks an analysis of the role of fine-tuning. An ablation study on this would further strengthen the work.

Finally, I also found the summary of the method in the introduction to be vague, and it was difficult to comprehend. In general, this work could be strengthened by being more precise in its contributions and presentation.

---

> ### Author Rebuttal · Authors · 2025-07-31
>
> Thank you for carefully reviewing our manuscript and providing valuable suggestions. We have addressed each of the issues you raised and will incorporate revisions into our latest manuscript.
>
> > **Weakness 1:** Lack of contextualization
>
> > **Weakness 1.1: The authors claim that using labeled preference pairs is limited by annotation difficulty, yet POEM also depends on human rankings during fine-tuning.**
>
> **Answer:** We apologize for any misunderstanding caused by our unclear wording. What we intended to convey is that other methods require large amounts of annotated preference pairs to train reward models, which makes them difficult to scale. As reported in **our response to Reviewer 4MF9’s Question 2**, existing reward modeling methods annotate a substantial number of preference pairs to achieve better performance. For example, training WorldPM-72B [1] and Skywork-Reward-8B [2] requires **15M** and **40M** labeled preference pairs, respectively, whereas fine-tuning our POEM-1.8B only requires **0.15M** labeled data. The amount of labeled data needed to fine-tune the POEM model is much fewer than that required by existing SOTA reward models.
>
> Furthermore, to further improve performance, these traditional reward models would require even more annotated preference pairs. However, manual annotation is resource-intensive, while automated annotation may introduce bias.
> In contrast, POEM exhibits strong generalization ability and achieves scalability through pre-training. It does not require the massive amount of annotated data needed by traditional reward modeling methods.
> This is the key point we wanted to express.
>
> We will refine our statement and include a comparison of training data requirements between POEM and traditional reward modeling methods in the main text to help readers better understand our work.
>
> > **Weakness 1.2: The paper criticizes Bradley-Terry Loss for assuming absolute preferences, yet POEM uses a similar method with three samples, showing recurring inconsistencies.**
>
> **Answer:** We apologize for any confusion caused by our unclear statements. We would like to clarify that we are not criticizing the form of the BT framework itself, but rather arguing against the way current reward modeling methods use the BT framework to model **absolute preferences**. Specifically, these methods utilize the BT loss to model what constitutes good or bad preferences. However, the preference data used for training may not be able to capture a wide range of preferences. Moreover, conflicting criteria may lead to contradictory partial orderings within the same training preference data.
>
> In contrast, although POEM also employs the BT framework for training, it models **relative difference**. For example, in RLHF, relative difference can be understood as the gap between the training policy and the target policy, rather than making absolute judgments about individual policy.
>
> Therefore, in the pre-training phase, POEM trains the model to distinguish between policies, while in the fine-tuning phase, it learns to align with human perception of the differences between two policies. Through this approach, POEM avoids modeling absolute preferences directly. As a result, POEM can scale its capabilities through pre-training without requiring extensive annotation of fine-tuning data.
>
> We will refine our statements in our latest manuscript to more clearly articulate the limitations of traditional RMs.
>
> > **Weakness 1.3: Needs better context and discussion of limitations.**
>
> **Answer:** Thank you for your valuable suggestions regarding our manuscript. In addition to addressing the two specific concerns you mentioned, we will thoroughly review and further refine our manuscript. We will do our best to ensure that our contributions are clearly and effectively communicated to the readers.
>
> > **Weakness 2: I wonder if POEM’s success mainly comes from its reformulated loss. I may be missing something, so I’d appreciate the authors’ perspective.**
>
> **Answer:** Thank you for your valuable comments. As we mentioned in **our response to Weakness 1.2**, the success of POEM stems from a fundamental shift in its modeling objective (from absolute preference to relative difference).
>
> Traditional reward modeling methods aim to enable RMs to learn absolute preferences. These methods require continuous annotation of preference data in an attempt to comprehensively cover human preferences, which is inherently difficult to scale. In contrast, POEM is designed to learn relative difference. During pre-training, POEM encourages the model to distinguish between different policies; during fine-tuning, it trains the model to align with human perception. Both stages share the same goal: modeling relative difference.
>
> This shift in modeling objective makes unsupervised pre-training of the RM possible, similar to how pre-training is crucial for the success of LLMs. We can construct large-scale pre-training datasets for POEM without the need for manual annotation, continuously improving its performance. Our scaling experiments clearly show the scaling laws of POEM. Moreover, ablation studies further show that if the POEM model is only fine-tuned on annotated data without pre-training, its generalization ability drops significantly. This provides further evidence that pre-training brings strong generalization capabilities.
>
> > **Weakness 3: No analysis on policy set sensitivity.**
>
> **Answer:** Thank you for your valuable suggestions. We agree that the choice of policies used to construct the pre-training dataset can affect the quality of the pre-training data, and consequently influence POEM’s capabilities. However, investigating this question requires substantial computational resources. As reported in Appendix C, training POEM-1.8B consumed significant computational resources: we used 320 NVIDIA H20 GPUs for approximately 57 hours. Therefore, due to budget constraints, we finally utilized all available policies (184 models) in practice.
>
> Fortunately, even without carefully selecting the policies for pre-training, POEM-1.8B demonstrated remarkable performance, showing strong effectiveness and generalization in both preference prediction evaluation and RLHF, significantly outperforming all SOTA baselines.
>
> In the future, we will ensure to invest more computational resources to better understand the impact of policy set selection on performance. We also plan to train larger models to further advance POEM’s performance.
>
> > **Weakness 4: The paper justifies pretraining but not fine-tuning in POEM’s two-stage training. An ablation study on fine-tuning would be valuable.**
>
> **Answer:** We conduct an ablation study by evaluating the preference prediction performance of the pre-trained-only model, as shown in the table below.
>
> |Model|pre-trained-only PDM-1.8B |POEM-1.8B|
> |---|---|---|
> |Harmlessness|65.7%|74.2%|
> |Brainstorming|63.5%|74.8%|
> |Chat|67.8%|75.1%|
> |Role Playing|60.3%|70.6%|
> |NLP Tasks|70.6%|68.5%|
> |STEM|66.7%|79.8%|
> |Coding|59.2%|65.8%|
> |Creative Writing|64.5%|85.5%|
> |Reasoning|92.3%|69.2%|
> |Instruct Following|65.4%|73.1%|
> |Closed&Open QA|70.4%|80.3%|
> |Multilingual|73.1%|80.8%|
> |**Avg.**|*66.3%*|*74.0%*|
>
> We observe that, without alignment to human judgments, POEM-1.8B fails to achieve meaningful scores on preference evaluation. This is reasonable, as only after POEM’s judgments are aligned with human judgments can it effectively leverage the broad knowledge acquired during pre-training to achieve better performance on downstream tasks. This is similar to LLMs, where LLMs need to be fine-tuned with a small amount of labeled data to align with human instructions. In essence, fine-tuning enables POEM to distinguish between policies from a human perspective.
>
> Combined with the ablation study in our manuscript, which evaluates POEM without pre-training, it is clear that both stages of POEM play an irreplaceable role.
>
> > **Weakness 5: Introduction and contributions are vague and unclear.**
>
> **Answer:** Thank you for your valuable suggestions. We will add more details about our method in the introduction to help readers gain a better understanding of our approach early on.
>
> > **Question1-4: Some typo, notation, and expression issues**
>
> **Answer:** Thank you for your valuable comments! We have addressed the following points:
>
> **(1) Notation (Line 123):** We confirm that $\tau$ should refer to a single trajectory, not a set. We will revise the notation for clarity.
>
> **(2) Equation Citations:** Equation 1 follows the RL objective in Ouyang et al. [3], and Equation 2 is based on the closed-form solution from Rafailov et al. [4]. We will add the appropriate citations.
>
> **(3) Complementary Criterion (Line 140):** Good catch. We will revise the phrasing and have confirmed that this does not impact the validity of our subsequent arguments.
>
> **(4) Notation in Eq. 6:** $\sigma$ denotes the sigmoid function. We will clarify this explicitly in the revised manuscript.
>
> Moreover, we will carefully review the entire paper to ensure that all concepts are clearly explained, further helping all readers to better understand our work.
>
> Finally, we would like to sincerely thank the reviewer for all the valuable suggestions. We will make sure to incorporate all of the reviewer’s suggestions into our latest manuscript. If you have any further comments, please do let us know, and we will do our best to address them.
>
> **References:**
>
> [1] Wang, B. et al. (2025). WorldPM: Scaling Human Preference Modeling. arXiv.
>
> [2] Liu, C. Y. et al. (2025). Skywork-Reward-V2: Scaling Preference Data Curation via Human-AI Synergy. arXiv.
>
> [3] Ouyang, L. et al. (2022). Training language models to follow instructions with human feedback. NeurIPS, 35, 27730-44.
>
> [4] Rafailov, R. et al. (2023). Direct preference optimization: Your language model is secretly a reward model. NeurIPS, 36, 53728-41.

---

> > ### Comment · Area_Chair_yRr7 · 2025-08-05
> >
> > Reviewer wgBV, can you please check whether the author's rebuttal addresses your concerns?

---

> > ### Comment · Reviewer_wgBV · 2025-08-07
> >
> > I thank the authors for their detailed rebuttal on my concerns and questions. The rebuttal is responsive to my main points. While I share reviewer cmE2’s view that the reliance on a reference trajectory remains a notable limitation ( a point the authors themselves rightfully acknowledge), the empirical performance of the proposed method is promising. In light of this, I am raising my score.

---

> > > ### Author Response · Authors · 2025-08-07
> > > **Response to Reviewer**
> > >
> > > Thank you for your valuable response. We’re glad that our rebuttal addressed your main points.
> > >
> > > We hope that POEM can inspire further research and exploration in the community on this new reward modeling paradigm and the reward model pretraining.
> > >
> > > Once again, we sincerely appreciate your recognition of our work and your decision to raise the score!

---

### Official Review · Reviewer_kicK · 2025-07-02

**Clarity:** 4
**Significance:** 3
**Originality:** 4
**Rating:** 5
**Confidence:** 4

**Summary:**

The paper introduces Policy Differentiation Modeling (POEM), a two-stage framework for training LLM reward models (RMs) without relying on the conventional pair-wise ranking loss.

- Pre-training: POEM is first trained with a contrastive objective that encourages it to differentiate trajectories generated from different policies, conditioned on the same prompt.

- Fine-tuning: Given a prompt and three outputs ordered by quality, the model receives the top-ranked output as a reference and is optimized so that the score for the second-best trajectory exceeds that of the third-best.

A 1.8-billion-parameter instance, POEM-1.8B, is shown to on average surpass baseline reward models both in human preference studies and when used to drive RLHF training.

**Questions:**

1. Could you provide empirical results—e.g., a plot of POEM performance when using the trajectory from the current prompt as reference versus using a trajectory of the same reference policy but from a different prompt—to substantiate the claim you make in Section 5?
2. Was POEM initialised from an existing pretrained LLM (and if so, which architecture and checkpoint), or trained from scratch?

**Ethical Concerns:**

["NO or VERY MINOR ethics concerns only"]

**Limitations:**

The authors have adequately addressed the limitations.

**Paper Formatting Concerns:**

I have no concerns regarding paper formatting.

**Quality:**

4

**Strengths And Weaknesses:**

### Strengths
- The method is rigorously described, with clear objectives, training procedures and hyperparameter settings.
- Writing is concise and well-structured.
- The results demonstrate that explicitly modelling policy differentiation can result in reward models that yield better RLHF policies.

### Weaknesses
- Dependence on reference trajecories from top-performing LLMs.
- The paper claims that “trajectories derived from other prompts may also effectively characterize policy behavior"  just as well as using the reference trajectory for the current prompt (Section 5) but does not include quantitative evidence or error bars demonstrating robustness to reference quality.

---

> ### Author Rebuttal · Authors · 2025-07-31
>
> Thank you very much for your recognition of our work and for providing valuable suggestions. We have addressed each of the issues you raised and will incorporate the corresponding revisions into our latest manuscript.
>
> > **Weakness 1: Dependence on reference trajectories from top-performing LLMs.**
>
> **Answer:** Thank you very much for your valuable comments. We provide a comprehensive discussion on why this limitation may not be a significant issue for several reasons, as detailed in **our response to Reviewer cmE2’s Weakness 1.** We hope this further discussion addresses your concern.
> If you have any further questions or would like to discuss in more detail, please feel free to contact me.
>
> > **Weakness 2 & Question 1: The paper claims that “trajectories derived from other prompts may also effectively characterize policy behavior" just as well as using the reference trajectory for the current prompt (Section 5) but does not include quantitative evidence or error bars demonstrating robustness to reference quality. Could you provide empirical results—e.g., a plot of POEM performance when using the trajectory from the current prompt as reference versus using a trajectory of the same reference policy but from a different prompt—to substantiate the claim you make in Section 5?**
>
> **Answer:** Thank you for your insightful question. We would like to clarify that our discussion on this topic is presented in Section 5 (Limitations and Future Work), where we mainly aim to share our latest observations from the case study. This does not include systematic quantitative experiments, and we would leave them as future work. We appreciate the opportunity to further elaborate here.
>
> The motivation behind using trajectories from different prompts is to alleviate POEM’s reliance on prompt-specific references. POEM is designed to assign reward values based on the consistency between two different policies. In our preliminary exploration, we observed that the reference and the trajectory can correspond to different prompts, as long as they share commonalities.
>
> Specifically, POEM was originally intended to score based on `(Prompt, Reference, Trajectory)`. However, we found that it can also score based on `(Prompt A, Reference, Prompt B, Trajectory)`, where "Reference" represents Policy A's answer to Prompt A, and "Trajectory" represents Policy B's answer to Prompt B. Formally, the input format changes from
> `<|bos|> Prompt <|split_token|> Reference <|split_token|> Trajectory <|eos|>`
> to
> `<|bos|> Prompt A <|split_token|> Reference <|split_token|> Prompt B <|split_token|> Trajectory <|eos|>`.
>
>
> We illustrate this with the following case study:
>
> ```
> Prompt A: "All birds have wings. Tweety is a bird. Question: Does Tweety have wings?"
>
> Reference: "Yes. Tweety has wings."
>
> Prompt B: "All teachers work at schools. Mr. Smith is a teacher. Question: What can we conclude?"
>
> Trajectory 1: "Mr. Smith works at a school." # Correct
> => Reward: -9.0390625
>
> Trajectory 2: "People who work at schools are teachers." # Incorrect
> => Reward: -11.015625
> ```
> In this case, both Prompt A and Prompt B are classic deductive syllogisms, each requiring reasoning from two premises (one general and one specific) to reach a necessary conclusion.
> Although the question types differ (Prompt A is yes/no, while Prompt B is open-ended), the underlying logic is similar. For Prompt B, Trajectory 1 provides the correct conclusion, whereas Trajectory 2 incorrectly reverses the premise.
>
> Results show that, when given Prompt A and its reference, POEM can assign a higher reward to the correct answer to Prompt B. This suggests that POEM has the potential to generalize scoring across prompts by referring to similar question-answer pairs. Such a capability could help reduce the annotation burden of generating a reference for every individual prompt.
>
> We will update Section 5 (Limitations and Future Work) to clarify the current scope of our findings. In future work, we plan to further investigate this finding and systematically evaluate POEM’s performance when using references and trajectories from the same versus different prompts.
>
> > **Question 2: Was POEM initialised from an existing pretrained LLM (and if so, which architecture and checkpoint), or trained from scratch?**
>
> **Answer:** The pre-training phase of POEM is initialized from an existing pretrained LLM. Specifically, POEM is pre-trained starting from the InternLM2 series InternLM2.5-1.8b-chat model [1], as presented in Appendix C.1.2 (Implementations, Line 837) of our manuscript, due to space limitations in the main text. We will **move this information to the main text** to provide readers with a clearer understanding of our training setup.
>
> Finally, we sincerely thank the reviewer for their recognition and for all the valuable suggestions provided for our manuscript. We will make sure to incorporate all of the suggestions into the next version of our manuscript. If you have any further comments, please do let us know, and we will do our best to address them.
>
> **References:**
>
> [1] Cai, Zheng, Maosong Cao, Haojiong Chen, Kai Chen, Keyu Chen, Xin Chen, Xun Chen et al. "InternLM2 Technical Report." CoRR (2024).

---

> > ### Comment · Reviewer_kicK · 2025-08-05
> >
> > Thank you for your detailed response and clarifications. I appreciate the effort you put into addressing my comments. I am satisfied with your reply and will maintain my current positive evaluation of the paper.

---

> > > ### Author Response · Authors · 2025-08-05
> > > **Response to Reviewer**
> > >
> > > We sincerely appreciate your insightful comments once again. We are glad that our responses have addressed your comments and that you are satisfied with them.
> > >
> > > Thank you very much for your continued support and positive evaluation of our work!

---

### Official Review · Reviewer_cmE2 · 2025-07-02

**Clarity:** 3
**Significance:** 3
**Originality:** 3
**Rating:** 4
**Confidence:** 2

**Summary:**

This paper redefines reward modeling not as predicting absolute preferences, but as a 'policy discriminator' that quantifies differences between policies. Based on this perspective, it proposes POEM (POlicy DiffErentiation Modeling), a scalable and criterion-agnostic reward model pre-training approach. POEM learns reward signals by distinguishing between various policy behaviors and is then fine-tuned with a small amount of human-labeled data to align with human preferences.

**Questions:**

- The authors mention that they report the best results between reference-free and reference-included evaluations(lines237-238), but it would be more informative to separate the two and compare performance explicitly based on the presence or absence of the reference.
- The impact of the pre-training stage on performance does not appear to be very large. As shown in Table 9, the average difference between POEM and the model w/o pre-training is only about 1.4 points. This raises the question of whether meaningful performance can still be achieved without pre-training. It would be helpful if the benefits of the pre-training stage were explained in more depth.
- In scenarios where a reference is available, alternative methods could compute rewards based on similarity to the reference (e.g., embedding similarity). How would such approaches perform compared to POEM, and what are the fundamental differences between them?

**Ethical Concerns:**

["NO or VERY MINOR ethics concerns only"]

**Final Justification:**

Many of my questions were resolved through the author's response. Although I still believe that using references is not entirely fair, I acknowledge that POEM demonstrates superior performance compared to other methods that also use references. Therefore, I am raising my score from 2 (reject) to 4 (borderline accept).

**Limitations:**

Same as mentioned in the weakness section

**Paper Formatting Concerns:**

Some of the figures and tables discussed in the main text—such as Table 9, 10, and 11—are placed in the appendix. If these are central to the paper’s main arguments or claims, it would be more appropriate to include them in the main body of the paper rather than relegating them to the appendix.

**Quality:**

3

**Strengths And Weaknesses:**

- Strength
    - The paper is structurally well-organized and easy to follow.
    - It achieves strong performance even with relatively little human-labeled data.
    - It demonstrates high efficiency relative to model size and shows superior empirical performance compared to baselines.
- Weakness
    - The most significant weakness is that the reward model requires a reference during reward generation. This contradicts the typical assumption in evaluation, where the quality of references is unknown. As such, assuming access to high-quality references makes it difficult to consider the comparison with baselines truly fair and poses practical limitations for real-world deployment.
    - In other words, the reward model is heavily dependent on the reference, and its performance cannot be guaranteed when the reference is sub-optimal or hard to obtain.

---

> ### Author Rebuttal · Authors · 2025-07-30
>
> Thank you very much for providing valuable suggestions. We have addressed each of the issues you raised and will make revisions to our latest manuscript.
>
> > **Weakness 1: The most significant weakness is that the reward model requires a reference during reward generation.**
>
> **Answer:** Thank you for raising this concern. Compared to traditional RM methods, POEM does require the introduction of additional references, which we have discussed in the limitations section. We hope the following further discussion will address your concern:
>
> (1) In our RL experiments, POEM significantly outperforms conventional reward models. Given POEM’s strong performance, generalization, and scalability, the requirement of references is not expected to substantially limit its broader adoption and future development. **One supporting example** is Reinforcement Learning from Verifiable Reward (RLVR), a widely adopted post-training paradigm for verifiable tasks. In RLVR, each prompt also requires a corresponding reference answer, enabling the verifier to assess the correctness of the trajectory. For example, in mathematical reasoning tasks, it is common practice to use rules to compare the generated solution with the reference answer for reward assignment. RLVR has become popular precisely because its reward signals are sufficiently reliable, and the need for references has not hindered its widespread adoption.
> POEM can be regarded as **a general-domain verifier**. Experimental results show that POEM has generalization and scalability, which provides stable and accurate reward signals in RL, significantly outperforming SOTA reward model baselines.
>
> (2) The need for references can be addressed through **open-source LLM instruction tuning datasets and data synthesis**. Specifically, the reference requirement for POEM is consistent with that for instruction tuning in LLMs, where each prompt also requires a reference. There are now many high-quality, open-source instruction tuning datasets that can serve as references for POEM.
> For references that are difficult to annotate, they can be synthesized using more powerful LLMs. In fact, the references used in our RLHF experiments were also synthesized (as indicated in Line 906). The results still show that POEM can achieve significantly better performance than all SOTA RM baselines, even without requiring a perfect reference.
>
> In summary, although POEM requires references during reward generation, this limitation may not be a major concern due to the following factors: **1.** the need for references is consistent with the verifier in the widely adopted RLVR paradigm, and POEM can be viewed as extending the verifier approach from verifable tasks to more general domains; **2.** its data requirements are aligned with those for instruction tuning in LLMs; and **3.** our experiments demonstrate that synthesizing references is feasible. Therefore, the need for references may not be a major concern, especially given the significant advantages POEM offers over traditional RMs.
>
>
> > **Weakness 2: Assuming access to high-quality references makes it difficult to consider the comparison with baselines truly fair and poses practical limitations for real-world deployment.**
>
> **Answer:** We would like to clarify that we have tried our best to ensure a fair comparison with existing SOTA reward model baselines. As mentioned in the Evaluation Setup section (line 237), to ensure fairness, we evaluate these baselines using two approaches, i.e., scoring with and without the reference. For each baseline, we report **the best results** across these two settings. Additionally, as discussed in our response to Weakness 1 and in the manuscript (line 906), references can be synthesized from top-performing LLMs.
>
> The results show that, even when using synthesized references as input, POEM-1.8B outperforms these SOTA baselines in all settings, while using significantly fewer parameters. POEM-1.8B also exhibits stronger generalization capabilities. We hope these clarifications address your concerns.
>
>
> > **Question 1: The authors mention that they report the best results between reference-free and reference-included evaluations(lines237-238), but it would be more informative to separate the two and compare performance explicitly based on the presence or absence of the reference.**
>
>
> **Answer:** Following your suggestion, I have reported the performance of the two SOTA baselines in these two settings on the preference prediction evaluation, as shown in the table below.
>
> |Category|POEM-1.8B|InternLM2-Reward-20B w/o Ref|InternLM2-Reward-20B w/ Ref|Skywork-Reward-27B w/o Ref|Skywork-Reward-27B w/ Ref|OpenAI Text-Embedding-3-Large|
> |---|---|---|---|---|---|---|
> |Harmlessness|74.2%|66.8%|64.8%|75.9%|76.1%|66.1%|
> |Brainstorming|74.8%|74.8%|72.3%|73.6%|68.6%|61.8%|
> |Chat|75.1%|69.5%|62.1%|72.3%|67.2%|71.1%|
> |Role Playing|70.6%|75.0%|67.7%|70.6%|69.1%|70.6%|
> |NLP Tasks|68.5%|70.6%|67.8%|71.3%|61.9%|62.8%|
> |STEM|79.8%|52.4%|60.7%|54.8%|52.4%|72.9%|
> |Coding|65.8%|63.2%|54.0%|60.5%|68.4%|47.1%|
> |Creative Writing|85.5%|51.3%|60.5%|56.6%|60.5%|78.1%|
> |Reasoning|69.2%|53.9%|46.2%|53.9%|46.2%|77.8%|
> |Instruct Following|73.1%|65.4%|69.2%|69.2%|73.1%|50.0%|
> |Closed&Open QA|80.3%|56.3%|52.1%|62.0%|56.3%|57.1%|
> |Multilingual|80.8%|69.2%|64.0%|76.9%|60.0%|64.0%|
> |**Avg.**|*74.0%*|*66.7%*|*64.5%*|*72.4%*|*70.6%*|*65.5%*|
>
> The experimental results indicate that, for most tasks, providing a reference actually leads to decreased preference prediction performance for these traditional reward model baselines. This may be because these traditional reward models are optimized to model absolute preferences (i.e., determining what is good or bad), rather than relative difference, which focuses on comparing the quality of two policies.
>
> > **Question 2: The impact of the pre-training stage on performance does not appear to be very large. As shown in Table 9, the average difference between POEM and the model w/o pre-training is only about 1.4 points. It would be helpful if the benefits of the pre-training stage were explained in more depth.**
>
>
> **Answer:** Thank you for your valuable question. Overall, pre-training enables POEM to achieve **a strong generalization ability** in RL practice. In fact, Table 9 presents a comparison between POEM with and without the pre-training phase in preference prediction evaluation. While preference prediction is the most commonly used and convenient setting for validating reward model effectiveness, it is not considered the “gold standard” for evaluating the true effectiveness and generalization of reward models, in fact, the performance of reward models in RLHF provides a more reliable evaluation method [1].
>
> In Table 10 of our manuscript, we compare the effectiveness of POEM in RLHF with and without the pre-training phase. The results show that pre-trained POEM outperforms its non-pre-trained counterpart on nearly all tasks (**19 out of 20**), providing strong evidence that pre-training significantly enhances the generalization ability.
> Moreover, we explored the scaling laws during the pre-training phase, as shown in Figure 4. The results indicate that POEM has the potential to achieve even stronger performance through scaling.
>
> In summary, POEM leverages unsupervised pre-training on large-scale data to learn to distinguish differences between policies. The pre-training phase is indispensable, serving as the foundation for POEM’s strong generalization ability and scalability.
>
>
> > **Question 3: In scenarios where a reference is available, alternative methods could compute rewards based on similarity to the reference (e.g., embedding similarity). How would such approaches perform compared to POEM, and what are the fundamental differences between them?**
>
> Answer: We conduct experiments comparing POEM-1.8B with the SOTA embedding model on preference prediction evaluation. For the baseline, we calculate the similarity between the embeddings of the reference and the trajectory as the reward value. The experimental results are shown in the table in response to Question 1. The results show that, even when using OpenAI’s most powerful embedding model, our method significantly outperforms it in preference prediction.
>
> There is a **fundamental difference** between our method and these similarity-based approaches. Although POEM also takes two samples as input, it measures the consistency between two policies represented by these two samples, rather than just their surface-level similarity. By focusing on this objective, POEM can effectively evaluate the consistency between the training policy and the target policy in RL, assigning more meaningful rewards.
>
>
> > **Paper Formatting Concerns: Some of the figures and tables discussed in the main text—such as Table 9, 10, and 11—are placed in the appendix.**
>
>
> **Answer:** Thank you for your suggestions! In fact, Table 1 in the main text is a simplified version of Tables 9, 10, and 11, which aggregate the benchmarks according to their evaluation capabilities. We will add this explanation to the caption of Table 1 to help readers better understand the relationship among these tables. Additionally, if our work is accepted, we will move these tables into the main text to further improve the clarity of our paper.
>
> Finally, we sincerely thank the reviewer for all the valuable suggestions provided for our manuscript. We will carefully revise our manuscript and incorporate these experiments into our latest version. If you have any further questions or suggestions, please feel free to reach out to us.
>
>
> **References:**
>
> [1] Zhou, E., Zheng, G., Wang, B., Xi, Z., Dou, S., Bao, R., ... & Huang, X. RMB: Comprehensively benchmarking reward models in LLM alignment. In The Thirteenth International Conference on Learning Representations (ICLR 2025).

---

> > ### Comment · Reviewer_cmE2 · 2025-08-05
> >
> > Many of my questions were resolved through the author's response. Although I still believe that using references is not entirely fair, I acknowledge that POEM demonstrates superior performance compared to other methods that also use references. Therefore, I am raising my score from 2 (reject) to 4 (borderline accept).

---

> ### Author Response · Authors · 2025-08-05
> **Response to Reviewer**
>
> Thank you for your response. We are glad that our rebuttal has resolved many of your questions, and we sincerely appreciate your recognition of our work and your decision to raise the score.
>
> We agree that achieving an entirely fair comparison is inherently challenging, as POEM introduces a novel and scalable reward modeling paradigm that fundamentally differs from traditional approaches. Nevertheless, we have sought to ensure fair and rigorous comparisons with SOTA reward model baselines in our experimental setup.
>
> Thank you again for your valuable comments and for acknowledging our efforts!

---

### Official Review · Reviewer_4MF9 · 2025-07-02

**Clarity:** 3
**Significance:** 4
**Originality:** 3
**Rating:** 5
**Confidence:** 4

**Summary:**

The paper presents a novel approach for aligning large language models (LLMs) via reinforcement learning using a discriminative reward model. The central idea is to train a discriminator to distinguish between high- and low-quality responses, which then serves as a reward signal for RL-based fine-tuning. The method is theoretically grounded and extensively evaluated across a broad spectrum of tasks, model scales, and baselines. The proposed discriminator-based reward model demonstrates improved scalability, robustness, and overall performance compared to prior approaches.

**Questions:**

Could the authors clarify the intuition behind the fine-tuning loss in Eq. 7? Specifically, what motivates replacing $(A_1, A_2 | B_1, B_2)$ with $AC | AB$? A brief intuitive explanation would enhance understanding.

What are the compute and data requirements for training the discriminator at scale compared to other alignment strategies (e.g., current preference models)? How does the scalability presented in 4.4 compare to the considered baseline approaches?

**Ethical Concerns:**

["NO or VERY MINOR ethics concerns only"]

**Final Justification:**

The rebuttal effectively addresses the previously raised questions and concerns, and demonstrates careful consideration of the feedback. In light of this, the initially positive assessment is maintained.

**Limitations:**

The paper presents a thoughtful discussion on limitations.

**Paper Formatting Concerns:**

None noted, though the addition of a brief preliminaries section to introduce notation and setup would enhance readability.

**Quality:**

4

**Strengths And Weaknesses:**

## Strengths

- Motivation & Novelty: The use of a discriminator as a reward model is well motivated, offering a scalable and principled alternative to traditional preference modeling.
- Theoretical Soundness: The approach is supported by a rigorous theoretical framework, enhancing its credibility.
- Comprehensive Evaluation: The paper provides extensive empirical validation, evaluating both the reward model itself and its downstream performance for RL-based fine-tuning. Evaluation spans a variety of model sizes, task domains, and baselines.
- Strong Results: The method consistently shows strong and often state-of-the-art results, indicating its practical effectiveness and generality.

## Weaknesses

- Preliminaries: The paper would benefit from a dedicated section summarizing the setup and notation, especially for readers less familiar with reward modeling in LLM alignment.
- Related Work: The method bears strong conceptual similarity to prior works such as *Generative Adversarial Imitation Learning* and *Discriminative Reward Co-Training*. A discussion on how this work extends or differentiates itself from these would help contextualize its contributions.

---

> ### Author Rebuttal · Authors · 2025-07-30
>
> Thank you very much for your recognition of our work and for providing valuable suggestions. We have addressed each of the issues you raised and will incorporate the corresponding revisions into our latest manuscript.
>
> > **Weakness 1: Preliminaries: The paper would benefit from a dedicated section summarizing the setup and notation, especially for readers less familiar with reward modeling in LLM alignment.**
>
> **Answer:** Thank you for your valuable suggestions. At the beginning of Section 3 (Methods), we will add an introduction to the experimental setup, notation, and conventional reward modeling methods to help readers better understand alignment and the contributions of our work.
>
> > **Weakness 2: Related Work: The method bears strong conceptual similarity to prior works such as Generative Adversarial Imitation Learning and Discriminative Reward Co-Training. A discussion on how this work extends or differentiates itself from these would help contextualize its contributions.**
>
> **Answer:** Thank you for your insightful suggestion. We have carefully reviewed the two papers provided by the reviewer. Both GAIL [1] and DIRECT [2] train a discriminator that distinguishes between trajectories generated by the current policy and those from high-quality trajectories, using the discriminator's output as a reward signal. In GAIL, expert demonstrations are used as high-quality trajectories, whereas DIRECT stores historically high-return action sequences as high-quality trajectories.
> These works primarily focus on enabling the training policy to perform similarly to an expert policy within a single task. In contrast, our work investigates whether reward models in LLM post-training can adopt a pre-training paradigm and exhibit strong generalization capabilities, similar to LLMs.
>
> Specifically, we explore how to construct unsupervised data to pre-train a policy discriminator, thereby redefining the reward function in LLMs. We further show its scaling laws and demonstrate its generalization and scalability through extensive experiments. Moreover, POEM can be fine-tuned with a small amount of labeled data to better adapt to downstream tasks, analogous to SFT in LLMs.
>
> In summary, our work differs significantly from previous studies in terms of training methodology, data synthesis paradigm, model generalization and scalability, as well as application scenarios.
> We will incorporate these points into our latest manuscript.
> Furthermore, we will thoroughly review related works that may share conceptual similarities with POEM and further expand Section 2 (Related Work) to help readers better understand our contributions.
>
> > **Question 1: Could the authors clarify the intuition behind the fine-tuning loss in Eq. 7? Specifically, what motivates replacing $(A_1, A_2 | B_1, B_2)$ with $AC | AB$? A brief intuitive explanation would enhance understanding.**
>
> **Answer:** POEM consists of two stages, i.e., the pre-training phase and the supervised fine-tuning phase. The goal of the first stage is to pre-train the reward model to distinguish between different policies. However, at this stage, the reward model is not yet aligned with human judgment; specifically, it does not capture the extent of differences between policy distributions as perceived by humans, nor does it reflect whether humans consider two policies to be different. Therefore, it is necessary to further fine-tune the reward model on a small amount of labeled data to achieve alignment with human judgment.
>
> Another intuitive explanation can be drawn from the training paradigm of LLMs. Although pre-trained LLMs acquire extensive knowledge, they are not directly suitable for downstream tasks. Instead, they require fine-tuning with a small amount of labeled data to learn to follow human instructions and align their language behavior with human expectations.
> We will refine our manuscript and incorporate these intuitive explanations to help readers better understand the purpose of the fine-tuning stage in POEM.
>
>
> > **Question 2: What are the compute and data requirements for training the discriminator at scale compared to other alignment strategies (e.g., current preference models)?**
>
> **Answer:** We compared our method with SOTA baselines in terms of compute and data requirements for training, as summarized in the table below. Missing values in the table indicate that the corresponding numbers were not disclosed by the respective methods.
>
> | Model  | Pre-training Data Size | Labeled Data Size  | Pre-training Compute Consumption   | Supervised Fine-tuning Compute Consumption   |
> |---------|--------|----|-------|--------|
> | WorldPM-72B [3]  | - | ~15M samples (15,000K)   | -   | -   |
> | Skywork-Reward-8B [4]   | -  | ~40M samples (40,000K)  | 64 NVIDIA H800 GPUs  | 64 NVIDIA H800 GPUs |
> | POEM-1.8B (Ours)  | ~0.94T tokens | ~150K samples | 320 NVIDIA H20 GPUs for ~57 hours| 16 NVIDIA H20 GPUs for ~0.5 hours   |
>
> Several key conclusions can be drawn from this table:
>
> (1) Our method introduces a pre-training phase, which requires additional compute resources. However, this phase is unsupervised and does not require labeled data, similar to the pre-training stage of LLMs.
> (2) **The amount of labeled data required by our method is significantly lower than that of traditional reward modeling approaches.** Specifically, we only need 150K labeled samples for supervised fine-tuning, whereas other methods typically require millions of labeled samples, which substantially increases annotation costs.
> (3) Moreover, our model has only 1.8B parameters, making it much more accessible for users to fine-tune POEM-1.8B compared to these larger models.
>
> In summary, although POEM consumes more compute during the pre-training phase, users can fine-tune POEM-1.8B with significantly less computational and annotation resources to obtain a more powerful reward model compared to other baselines.
>
> Furthermore, after the anonymity period ends, **we will open-source all our models, including the pre-trained model, SFT model, and training framework, to minimize user adoption costs. Users can directly utilize the SFT model or further fine-tune the reward model with a small amount of domain-specific data to achieve enhanced performance.**
>
>
> > **Question 3: How does the scalability presented in 4.4 compare to the considered baseline approaches?**
>
> **Answer:** Our method shows significantly better scalability compared to SOTA reward model baselines. As discussed in Section 1 (Introduction, line 26) and Section 2 (Related Work, line 90) of our manuscript, these baseline methods require the continuous annotation of a large number of preference pairs to improve performance. Consequently, they lack the scalability.
> In contrast, the pre-training phase of POEM is unsupervised and does not require labeled data, which ensures excellent scalability. During the fine-tuning phase, we only use a substantially smaller amount of labeled data than baseline methods to fine-tune the pre-trained model, yet still achieve stronger performance (as shown in experiments of our manuscript).
>
> Moreover, our scaling laws experiments (Figure 4) show a predictable decrease in validation loss as model size or computational resources increase. Furthermore, Tables 1, 5, 6, 7, and 8 demonstrate that POEM exhibits significantly better generalization and effectiveness compared to other SOTA baselines.
>
> In summary, these experimental results clearly indicate that our method offers substantially superior scalability.
>
> Finally, we sincerely thank the reviewer for their recognition and for all the valuable suggestions provided for our manuscript. We will make sure to incorporate all of the suggestions into the next version of our manuscript. If you have any further comments, please do let us know, and we will do our best to address them.
>
>
> **References:**
>
> [1] Ho, Jonathan, and Stefano Ermon. "Generative adversarial imitation learning." Advances in neural information processing systems 29 (2016).
>
> [2] Altmann, P., Ritz, F., Zorn, M. et al. Discriminative reward co-training. Neural Comput & Applic (2024). https://doi.org/10.1007/s00521-024-10512-8
>
> [3] Wang, B., Lin, R., Lu, K., Yu, L., Zhang, Z., Huang, F., ... & Lin, J. (2025). WorldPM: Scaling Human Preference Modeling. arXiv preprint arXiv:2505.10527.
>
> [4] Liu, C. Y., Zeng, L., Xiao, Y., He, J., Liu, J., Wang, C., ... & Zhou, Y. (2025). Skywork-Reward-V2: Scaling Preference Data Curation via Human-AI Synergy. arXiv preprint arXiv:2507.01352.

---

> > ### Comment · Area_Chair_yRr7 · 2025-08-05
> >
> > Reviewer 4MF9, can you please check whether the author's rebuttal addresses your concerns?

---

> > ### Comment · Reviewer_4MF9 · 2025-08-05
> >
> > Thank you for your detailed rebuttal addressing my concerns and responding to my open questions. I will maintain my already positive score.

---

> > > ### Author Response · Authors · 2025-08-05
> > > **Response to Reviewer**
> > >
> > > Thank you for your response! We are glad that our rebuttal has addressed your concerns and questions. We also greatly appreciate your support and recognition of our work!

---

### Note · Authors · 2025-08-12

We express our sincere gratitude to all reviewers for their valuable feedback, active engagement, and final recognition of our work.
We also deeply value your thoughtful evaluations, which highlight several strengths of our work:

1. Well-motivated and novel perspective on reward modeling (Reviewer 4MF9, wgBV)
2. Rigorous theoretical framework and comprehensive empirical validation (Reviewer 4MF9, kicK)
3. Strong empirical results and scalability, consistently outperforming existing baselines (Reviewer 4MF9, cmE2, kicK)
4. Clear and structured presentation (Reviewer cmE2, kicK)

A major concern shared by reviewers is the reliance on references during reward generation. Additionally, reviewers suggested adding a dedicated section for preliminaries, expanding the discussion of related works, and improving method clarity, especially for readers less familiar with LLM reward modeling.

We provide detailed responses to these concerns in the rebuttal, which receives substantial recognition from the reviewers.
Specifically, for the reliance on references:
- We acknowledge that POEM requires an additional reference, as discussed in our limitations section. In our rebuttal, we have explained in detail why this requirement does not limit POEM’s applicability, considering aspects including (1) empirical performance and generalization, (2) consistency with mainstream RLVR paradigms, and (3) the availability and synthesis of reference data.
- Moreover, the reliance on references also brings important advantages. Unlike traditional methods that focus on absolute preferences, POEM models the relative difference between policies. The use of references enables POEM to flexibly capture these differences, supporting scalable pre-training and avoiding the limitations of absolute preference modeling.
- Additionally, as suggested by Reviewer cmE2, we provide explicit comparisons in both reference-free and reference-included evaluation settings to ensure fairer assessment, where POEM consistently outperforms baselines.

We will refine our paper based on the rebuttal content and further enhance its clarity and presentation, enabling the wider community to more effectively understand and benefit from our work.

Finally, we greatly appreciate the reviewers’ recognition of POEM’s distinctive contribution and potential to advance reward modeling. We are confident that our work advances RM pre-training and scalable reward modeling, and provides a foundation for future exploration.

---

### Decision · Program_Chairs · 2025-09-17

**Decision:**

Accept (poster)

**Comment:**

This paper introduces Policy Differentiation Modeling (POEM), a novel and scalable pre-training framework for reward models. All reviewers were positive, highlighting the novelty of framing reward modeling as policy discrimination, the strong empirical results, and the performance achieved with significantly less labeled data than the baseline. A key initial concern shared by reviewers was the method's reliance on a reference policy during reward generation, which raised questions about practical deployment and fairness of comparisons. The authors provided a comprehensive rebuttal that effectively addressed this limitation by providing additional experiments, leading multiple reviewers to raise their scores. The final consensus is that this paper presents a technically solid and impactful contribution. The paper is a clear accept.